



# A global map of Local Climate Zones to support earth system modelling and urban scale environmental science

Matthias Demuzere[1], Jonas Kittner[1], Alberto Martilli[2], Gerald Mills[3], Christian Moede[1], Iain D. Stewart[4], Jasper van Vliet[5], and Benjamin Bechtel[1]

[1]Urban Climatology Group, Department of Geography, Ruhr-University Bochum, Bochum, Germany
[2]Environmental Department, CIEMAT, Spain
[3]School of Geography, University College Dublin, Dublin, Ireland
[4]Global Cities Institute, University of Toronto, Toronto, Ontario, Canada
[5]Institute for Environmental Studies, Vrije Universiteit Amsterdam, De Boelelaan 1085, 1081, HV, Amsterdam, the Netherlands

**Correspondence:** Matthias Demuzere (matthias.demuzere@rub.de)

**Abstract.** There is a scientific consensus on the need for spatially detailed information on urban landscapes at a global scale. This data can support a range of environmental services, as cities are acknowledged as places of intense resource consumption and waste generation and foci of population and infrastructure that are exposed to multiple hazards of natural and anthropogenic origin. In the face of climate change, urban data is also required to explore future urbanisation pathways and urban design strategies, in order to lock in long-term resilience and sustainability, protecting cities from future decisions that could
undermine their adaptability. To serve this purpose, we present a 100m resolution global map of Local Climate Zones (LCZs), an universal urban typology that can distinguish urban areas on a holistic basis, accounting for the typical combination of micro-scale land-covers and associated physical properties. The global LCZ map, composed of 10 built and 7 natural land cover types, is generated by feeding an unprecedented amount of labelled training areas and earth observation imagery into
lightweight random forest models. Its quality is assessed using a bootstrap cross validation alongside a thematic benchmark for 150 selected functional urban areas using independent global and open-source data on surface cover, surface imperviousness, building height, and anthropogenic heat. As each LCZ type is associated with generic numerical descriptions of key urban canopy parameters that regulate atmospheric responses to urbanisation, the availability of this globally consistent and climate-relevant urban description is an important prerequisite for supporting model development and creating evidence-based
climate-sensitive urban planning policies. This dataset can be downloaded from http://doi.org/10.5281/zenodo.6364594 (Demuzere et al., 2022a).

## 1   Introduction

Cities are at the forefront of global climate change science owing to their emissions of greenhouse gases and their exposure to the projected hazards, such as sea-level rise and warming (IPCC, 2022). As a result, they are focus for mitigation and adaptation
policies and, as they have governance structures in place, are an ideal scale to affect change. The crucial role that cities can play in this arena is recognised at the international level: the new United Nations Agenda and the $11^{th}$ Sustainable Development



Goal focus on urban resilience, climate and environment sustainability of cities; two of the four challenges identified by the World Meteorological Organisation (WMO) World Weather Research Program are urban related: high-impact weather, including impacts in cities, and urbanisation; the Intergovernmental Panel on Climate Change Cities and Climate Change Scientific

Committee identified six research priorities for science to have a stronger role in urban policy and practice; and advocacy groups like C40[1] play an increasingly important role in achieving national emission targets and enhancing resilience (Creutzig et al., 2016; Bai et al., 2018; Masson et al., 2020).

Cities are simultaneously drivers of regional and local climate changes. The conversion of Earth's land surface to urban

areas is one of the most irreversible human impacts on the global ecosystem (Grimm et al., 2008; Reba and Seto, 2020). In addition to the many modifications to bio-, hydro-, and lithosphere (Seto et al., 2012; D'Amour et al., 2017; Liu et al., 2019; van Vliet, 2019; Zhang et al., 2019; McDonough et al., 2020), urbanisation affects energy demand (Creutzig et al., 2015; Güneralp et al., 2017), releases anthropogenic heat emissions and pollutants (Patella et al., 2018; Takane et al., 2019), and alters the urban climate (Oke et al., 2017). Current and future climate changes represent significant risks to urban populations and to

the natural and physical infrastructure systems of cities (Costello et al., 2009; UN, 2019; Wang et al., 2021). In this context, the WMO has advocated the development on integrated urban services (IUS) – using observations (remote and in situ) and models - that addresses the panoply of hazards that cities face and the needs of service providers, including emergency services, public health bodies, energy produces, urban designers and planners, etc. (Baklanov et al., 2018; Grimmond et al., 2020).

Despite their importance as a spatial nexus of climate drivers and of governance, cities are largely excluded from global climate science owing to their relatively small extent and our limited knowledge of their spatial structures. Earth System Models (ESMs) have only recently evolved to accommodate urban-scale landscapes, even though the parameters that are used by ESMs to these landscapes are limited in scope. At regional and urban scales, model developments use far more detailed parameters that include descriptions of the net impacts of buildings in creating a distinct urban canopy and boundary layers. While some

theoretical challenges remain, it is now possible to simulate urban effects on climate, between and above buildings at sub-urban scales (Barlow, 2014). Scientific advances will soon allow variable scale modelling that will incorporate the hierarchy of climate processes and impacts. However, the absence of suitable and universal global urban landscape data to inform these models represents a serious impediment to progress (Zhao et al., 2021; Hertwig et al., 2021). What is needed is a comprehensive database on cities globally that supports multi-scale modelling, provides a spatial framework for interpreting on-site

and remote measurements and allows the meaningful transfer of knowledge among and within cities (Rosenzweig et al., 2010; Hidalgo et al., 2018).

The critical data needed to support urban climate science includes information on urban form and functions. Measures of form include building density, street widths, building heights, construction materials and fraction of vegetated areas. These

attributes largely influences the local climate and the 'adaptation' capacity of a city (e. g. to ensure a comfortable thermal

---

[1]https://www.c40.org/cop26/





environment to its inhabitants). Urban functions describe the emissions of waste heat, materials and gases into the overlying atmosphere. Appropriate measures would include the anthropogenic heat flux (AHF) and $CO_2$ emissions. Form and function are correlated, for example: population density regulates energy consumption, and therefore the potential to mitigate global warming by reducing the greenhouse gas emissions; variations in building layout and heights moderates surface roughness and
contributes to the atmospheric dispersive conditions, and so the air quality (Martilli, 2014). Models are needed to assess the net benefits of climate-based interventions that may have unintended outcomes. For example, densely built and occupied cities (so called compact 15-minute cities) will reduce traffic, energy demand and $CO_2$ emissions, and in some cases improve air quality (Stone et al., 2007; McDonough et al., 2020; Williams et al., 2010), but will enhance warming and heat stress by reducing vegetative cover and sky view factor in the street canyons, and increase the spatial density of the anthropogenic heat(Demuzere
et al., 2014; Lai et al., 2019). Understanding how different urban forms interact with the atmosphere is key to redesigning cities, and, more importantly, plan future urbanisation. It is therefore essential to have information that differentiates between urban forms that can be used by atmospheric models to simulate the future climatic conditions and different urban form scenarios. Our objective here is to generate these data to support model evolution and stimulate research on multi-scale climate projections to manage urban risks.


Acquiring urban data at a global scale is not a trivial exercise owing to the operational definition of 'urban', the scattered extents of cities globally and their complex intra-urban geographies; for example, the Global Human Settlement Layer Urban Centres Database identifies over 13.000 settlements (Florczyk et al., 2019), while Li et al. (2020b) generated over 60.000 global urban boundaries. At the global scale, there are several datasets that identify the extent of contiguous urban areas, based on
built-up or impervious surface cover (Zhou et al., 2015; Corbane et al., 2017; Esch et al., 2017; Marconcini et al., 2020; Gong et al., 2020; Zhang et al., 2020a; Zhao et al., 2022) but none that provide intra-urban morphological details (green cover, built density, building heights, etc.) that are needed by scientists to generate the urban canopy parameters (UCPs) to run models and by urban policy-makers to make informed decisions based on analyses of risk. For many cities, relevant information may be gleaned from local sources that maintain municipal geographic databases (e.g. Biljecki et al., 2021), but these data vary
in terms of their quality, consistency, and accessibility, which limits their wider applicability (Zhu et al., 2019). The 100 m resolution global Local Climate Zone (LCZ) map presented here addresses this need for more detailed intra-urban data. This product is the outcome of more than a decade of research on how best to acquire, evaluate and deploy urban data in support of climate science (Stewart and Oke, 2012; Bechtel and Daneke, 2012; Ching et al., 2018).

The LCZ typology is currently the only universal classification that categorises urban landscapes using a scheme that identifies readily recognisable neighbourhood types based on their form and function, which modify the surface energy and water budgets. Critically, each LCZ type is linked to meaningful UCP value ranges that can be used for physically-based modelling (Stewart and Oke, 2012; Ching et al., 2019; Demuzere et al., 2020a). It goes beyond the urban mask and enables the assessment of the spatial impact of urban planning decisions that will alter UCPs and their climate outcomes. The LCZ scheme is distin-
guished from other LULC schemes by its focus on urban and rural landscape types, which can be described by any of the 17



classes in the scheme (Fig. 1). The scheme was originally designed to encourage climate change researchers to step away from their computers and get acquainted with the field sites that support their work, to capture the character of the urban landscape responsible for the urban heat island (UHI), and to ensure consistent reporting of metadata about the sites used to measure the heat island effect (Stewart, 2018; Stewart and Mills, 2021). The World Urban Database and Access Portal Tools (WUDAPT)

project has adopted the scheme in pursuit of its goal 'to capture consistent information on cities worldwide that can support urban weather, climate, hydrology and air quality modelling' Ching et al. (2018). The global LCZ product for the first time captures the intra-urban heterogeneity across the whole surface of the Earth, capturing cities of all sizes. It complements the LCZ maps for individual cities created by the WUDAPT community, the LCZ Generator [2] (Demuzere et al., 2021b) or that are available via other sources (e.g. Taubenböck et al., 2020; Zhu et al., 2022). In addition, as each LCZ type is associated with

generic numerical descriptions of key UCPs, the availability of this globally consistent and climate-relevant urban description is an important prerequisite to advance our capacity to assess climate risks at urban scales, and enable the development of fit-for-purpose climate mitigation and adaptation strategies.

---

[2]https://lcz-generator.rub.de/



## Built types

**Compact highrise**
**1** Dense mix of tall buildings to tens of stories. Few or no trees. Land cover mostly paved. Concrete, steel, stone, and glass construction materials.

**Compact midrise**
**2** Dense mix of midrise buildings (3–9 stories). Few or no trees. Land cover mostly paved. Stone, brick, tile, and concrete construction materials.

**Compact lowrise**
**3** Dense mix of lowrise buildings (1–3 stories). Few or no trees. Land cover mostly paved. Stone, brick, tile, and concrete construction materials.

**Open highrise**
**4** Open arrangement of tall buildings to tens of stories. Abundance of pervious land cover (low plants, trees). Concrete, steel, stone, and glass construction materials.

**Open midrise**
**5** Open arrangement of midrise buildings (3–9 stories). Abundance of pervious land cover (low plants, scattered trees). Concrete, steel, stone, and glass construction materials.

**Open lowrise**
**6** Open arrangement of lowrise buildings (1–3 stories). Abundance of pervious land cover (low plants, scattered trees). Wood, brick, stone, tile, and concrete construction materials.

**Lightweight lowrise**
**7** Dense mix of single-story buildings. Few or no trees. Land cover mostly hard-packed. Lightweight construction materials (e.g., wood, thatch, corrugated metal).

**Large lowrise**
**8** Open arrangement of large lowrise buildings (1–3 stories). Few or no trees. Land cover mostly paved. Steel, concrete, metal, and stone construction materials.

**Sparsely built**
**9** Sparse arrangement of small or medium-sized buildings in a natural setting. Abundance of pervious land cover (low plants, scattered trees).

**Heavy industry**
**10** Lowrise and midrise industrial structures (towers, tanks, stacks). Few or no trees. Land cover mostly paved or hard-packed. Metal, steel, and concrete construction materials.

## Land cover types

**Dense trees**
**A** Heavily wooded landscape of deciduous and/or evergreen trees. Land cover mostly pervious (low plants). Zone function is natural forest, tree cultivation or urban park.

**Scattered trees**
**B** Lightly wooded landscape of deciduous and/or evergreen trees. Land cover mostly pervious (low plants). Zone function is natural forest, tree cultivation, or urban park.

**Bush, scrub**
**C** Open arrangement of bushes, shrubs, and short, woody trees. Land cover mostly pervious (bare soil or sand). Zone function is natural scrubland or agriculture.

**Low plants**
**D** Featureless landscape of grass or herbaceous plants/crops. Few or no trees. Zone function is natural grassland, agriculture, or urban park.

**Bare rock or paved**
**E** Featureless landscape of rock or paved cover. Few or no trees or plants. Zone function is natural desert (rock) or urban transportation.

**Bare soil or sand**
**F** Featureless landscape of soil or sand cover. Few or no trees or plants. Zone function is natural desert or agriculture.

**Water**
**G** Large, open water bodies such as seas and lakes, or small bodies such as rivers, reservoirs, and lagoons.

### VARIABLE LAND COVER PROPERTIES

Variable or ephemeral land cover properties that change significantly with synoptic weather patterns, agricultural practices, and/or seasonal cycles.

**b. bare trees** — Leafless deciduous trees (e.g., winter). Increased sky view factor. Reduced albedo.

**s. snow cover** — Snow cover >10 cm in depth. Low admittance. High albedo.

**d. dry ground** — Parched soil. Low admittance. Large Bowen ratio. Increased albedo.

**w. wet ground** — Waterlogged soil. High admittance. Small Bowen ratio. Reduced albedo.

WUDAPT

**Figure 1.** Definitions of built (1-10) and land cover types (A-G) for the Local Climate Zone scheme (Stewart and Oke, 2012; Demuzere et al., 2020a).



## 2 Methods and Data

Whilst many LCZ mapping methodologies are currently available (see e.g. review by Jiang et al., 2021), the methodology for
the global LCZ map follows WUDAPT's default protocol that was launched by (Bechtel et al., 2015), sequentially improved
by Demuzere et al. (2019b, a, 2020a), ultimately leading to the LCZ Generator (Demuzere et al., 2021b), a web application
that makes single-city LCZ mapping fast and easy. The procedure requires labelled training areas, earth observation input data,
and a random forest model, discussed in-depth in Sections 2.1, 2.2 and 2.3 respectively. In addition, in line with previous
continental-scale LCZ mapping efforts by Demuzere et al. (2019b, 2020a), the quality of the resulting LCZ map is assessed in
two ways: 1) a traditional quality assessment using multiple accuracy metrics, and 2) a thematic benchmark, by translating the
LCZ map into its corresponding LCZ-based urban canopy parameters, and comparing these against (semi-)independent global
and open-source databases reflecting urban forms and functions (Section 2.4).

### 2.1 Training areas

Training areas (TAs) are LCZ-labelled polygons that represent typical examples of built or natural LCZs in a region of interest
(ROI). By design, they are compiled in a crowd-sourced manner, either by urban experts (Bechtel et al., 2015) or alternative
crowd-sourcing platforms such as MTurk (https://www.mturk.com) (Demuzere et al., 2020a; Xu et al., 2021) using good prac-
tice guidelines for digitising TAs (see Appendix A and Demuzere et al. (2021b)). While the LCZ maps created by individuals
are often of poor to moderate quality, The Human Influence Experiment (HUMINEX) (Bechtel et al., 2017; Verdonck et al.,
2019a) demonstrated large accuracy improvements (up to 20%) when multiple (poor to moderate quality) training datasets were
used together to create a single LCZ map. In the current study, TAs are compiled from multiple sources. First, well-trained
students assistants from the Ruhr-University Bochum produced TA sets for more than 100 global ROIs (labelled as RUB).
Second, archived TA (labelled as ARC) sets were collected from previous published research and collaborations, including the
samples hosted on the old WUDAPT portal (https://www.wudapt.org/the-wudapt-portal/). Finally, the RUB and ARC TA sets
are supplemented with the TA samples available from the LCZ Generator (Demuzere et al., 2021b) (labelled as GEN).


Before being used in the classification procedure, all TA sets are curated. First, all RUB and ARC training area sets are
submitted to the LCZ Generator: in case of multiple entries for one submission, only the best submission is retained. Also,
only TA samples mapped to LCZs with an overall accuracy greater than 50% are kept. Second, in case of duplicate regions
across the different sources, the following priority is used: RUB > ARC > GEN. Third, only the original seventeen LCZ
classes are kept, thereby removing non-standardized classes available in some of the samples, such as LCZ W - Wetlands
(Brousse et al., 2019, 2020b, a) and LCZ H - Agricultural greenhouses (Vandamme et al., 2019). Third, in order to maintain
computational efficiency, and to avoid redundancy and mixed spectral characteristics, the surface area of large polygons (>1.5
km$^2$) is reduced, and too small or too complex TA polygons are removed (Demuzere et al., 2021b). Finally, all pixels embedded
within the ROIs are assigned to urban ecoregions (ER), which are regional clusters based on climate, vegetation, and urban



topology (Schneider et al., 2010)). This is based on the finding of Demuzere et al. (2019b) that ERs can provide a basis for
       intelligent learning between cities and allow upscaling from individual cities to regional and global levels.

## 2.2    Earth observation input data

In addition to the TAs, one needs earth observation (EO) input data to feed the LCZ supervised random forest classifier
(Breiman, 2001; Bechtel et al., 2015). The 33 global earth observation input features used by default in the LCZ Generator (see

Table 2 in Demuzere et al., 2021b) serve as a baseline. However, some EO input features are updated or added. The original
       1 km global forest canopy height representative for 2005 (Simard et al., 2011) is replaced by the 30 m global forest canopy
       height dataset representative for 2019, developed by Potapov et al. (2021) through the integration of the Global Ecosystem
       Dynamics Investigation (GEDI) lidar instrument data (April–October 2019) and multi-temporal metrics derived from Landsat.
       Also the ALOS Digital Surface Model (DSM) data is updated to version 3.2, an improved version that reconsiders the format

in the high latitude area, auxiliary data, and processing method (Tadono et al., 2016). In addition, the SRTM Digital Eleva-
       tion Model (DEM) information is replaced by the MERIT DEM (Multi-Error-Removed Improved-Terrain Digital Elevation
       Model, Yamazaki et al. (2017)), from which also the slope and aspect are added. Because of the changes in the DSM and
       DEM, also the Canopy Height Model CHM (=DSM-DEM) data is updated. Building further upon the findings of Brousse et al.
       (2020a), Hay Chung et al. (2021) and Chen et al. (2021a), two more sets of input features are added, including: 1) Gray Level

Co-occurrence Matrix (GLCM) texture features (contrast, dissimilarity, inertia, sum average, and cluster shade) derived from
       PALSAR (Phased Array type L-band Synthetic Aperture Radar) for both HH and HV polarisations with a 4 by 4 kernel size
       (matching the LCZ 100 m spatial resolution), and 2) NANTLI, a Landsat 8 NDVI-adjusted (Normalized Difference Vegeta-
       tion Index) Night-Time Light Index based on VIIRS (Visible Infrared Imaging Radiometer Suite) data, analogous to EANTLI
       (Zhuo et al., 2015, 2018; Zhang et al., 2020a). See Appendix B for more details on these additional input features. Combined,

this results in a set of 46 earth observation input features, derived from Landsat 8 (16), Sentinel-1 (5), Sentinel-2 (8), PALSAR
       (10), VIIRS (1) and other sources (6).

## 2.3    Lightweight global random forest models

To date, the pixel-based LCZ mapping methods have used a wide variety of machine learning algorithms to classify LCZs (see

e.g. Section 3.1.4 in Jiang et al. (2021) for more details). Here, WUDAPT's initial and default (Bechtel et al., 2015) random
       forest classifier algorithm is used, building further upon the classification procedure of the LCZ Generator (Demuzere et al.,
       2021b), that uses Breiman's random forest implementation in Google's Earth Engine (EE), in combination with an automated
       cross-validation approach using 25 bootstraps (Breiman, 2001; Bechtel et al., 2015; Gorelick et al., 2017; Demuzere et al.,
       2019a, 2020a, 2021b). Yet given the sheer size of the classification problem (2+ million labels and 46 input features), two se-

quential pathways (Figure 2) are developed that lead to lightweight global random forest models that balance optimal learning
       with accuracy, computational feasibility and efficiency (Corbane et al., 2021).

**Figure 2.** Schematic representation of the sequential pathways to develop the global LCZ map.

In a first pathway, earth observation data are extracted from all input features and for all pixels embedded within the training area polygons. Then, using Python's random forest from the scikit-learn 0.24.2 package (Pedregosa et al., 2011) and the RF

parameters used in previous work by Demuzere et al. (2019a, 2020a, 2021b), a feature importance ranking is performed (Demuzere et al., 2019b, a), for the global TA set and 15 distinct TA sets stratified by urban ecoregion. Simultaneously, the quality of these random forest classifications are assessed (see Section 2.4 for more information) by bootstrapping the classification 25 times, for the global TA set and the 15 urban ecoregions, each time using a stratified (LCZ class) random TA sampling of 70 / 30% for training / testing. In addition, a hyperparameter tuning on EE's random forest parameters (e.g. number of trees,

maximum number of leaf nodes in each tree, minimum leaf population) was applied using Python's RandomSearchCV and GridSearchCV (Pedregosa et al., 2011) packages (not shown). But as the effect of different random forest parameters on the





overall accuracy was insignificant, the default random forest parameters were kept in pathway two.

The second pathway ingests the results from the first pathway to develop multiple lightweight global random forest models
within EE. First, the final input feature set is composed of the input features that belong at least 5 times (out of 16, reflecting
the global and 15 urban ecoregions) to the top 50% of most important features, obtained in pathway 1. Second, TA polygons
are sampled in a double cross-folding manner, using 5 seeds (random samples) and 10% of the resulting TA samples. This
is repeated 10 times, each time extracting a different 10% from the corresponding seed, resulting in 50 LCZ labels per pixel.
Note that this random sampling is balanced across LCZ labels and urban ecoregions, a sampling approach that meets the three
criteria as outlined by (Corbane et al., 2021; Xu et al., 2021): class balance, diversity, and representativeness. In a final step,
the modal LCZ class is selected as the final LCZ label, and the resulting global modal LCZ map is post-processed using the
morphological Gaussian filter described in (Demuzere et al., 2020a, 2021b). In addition, a probability layer is produced that
identifies how often the modal LCZ was chosen per pixel (e.g. a probability of 60% means that the modal LCZ class was
mapped 30 times out of 50 LCZ models).

## 2.4 Quality assessment and benchmarking

### 2.4.1 Traditional quality assessment

In order to assess the quality of the global LCZ map, the accuracy assessment from pathway 1 is repeated using the selected
earth observation features only. Also here, the pixel-based random forest classification is repeated 50 times (5 seeds × 10
distinct TA samples), and for each iteration, the TA sample is randomly split in a balanced manner (by urban ecoregion and
LCZ class) using 70 / 30% for training / testing. In order to avoid spatial autocorrelation that can lead to inflated accuracies,
the 'splitting the polygon pool' approach is used (Xu et al., 2021), in which the polygons (rather than the individual pixels)
are randomly sampled into 70/30 training / testing groups. The quality assessment is done using a range of well-accepted LCZ
accuracy metrics, including: overall accuracy (OA), overall accuracy for the urban LCZ classes only ($OA_u$), overall accuracy of
the built versus natural LCZ classes only ($OA_{bu}$), a weighted accuracy ($OA_w$), and the class-wise metric F1 (Chinchor, 1992;
Bechtel et al., 2017; Verdonck et al., 2017; Demuzere et al., 2019b, a; Bechtel et al., 2020). The overall accuracy denotes the
percentage of correctly classified pixels. $OA_u$ reflects the percentage of correctly classified pixels from the urban LCZ classes
only, and $OA_{bu}$ is the overall accuracy of the built versus natural LCZ classes only, ignoring their internal differentiation. The
weighted accuracy ($OA_w$) is obtained by applying weights to the confusion matrix and accounts for the (dis)similarity between
LCZ types (Bechtel et al., 2017, 2020). As such, confusion between dissimilar types (e.g. LCZs 1 A) is penalised more than
confusion between similar classes (e.g. LCZs 1 and 2). Finally, the class-wise accuracy is evaluated using the F1 metric, which
is a harmonic mean of the user's and producer's accuracy (Chinchor, 1992; Verdonck et al., 2017).



### 2.4.2 Thematic benchmark

A drawback of the traditional accuracy assessment is that only pixels within TA polygons are evaluated, and those outside are not quality-controlled. In addition, high overall accuracies do not automatically mean that the resulting LCZ map is correct, as e.g. an insufficient discrimination of LCZ types in the training sample can lead to an artificially high OA. To accommodate such limitations, the resulting LCZ map can be converted to its corresponding urban canopy parameters (Table 1), that are key in urban ecosystem processes (Stewart and Oke, 2012; Oke et al., 2017; Ching et al., 2018, 2019), and that offer an indirect thematic evaluation of the mapped LCZ quality. These UCP value ranges are not site-specific, but are designed to be universally applicable to all cities, since they are based on data gathered from a large sample of measurement studies, modelling studies, existing land-cover classifications, and urban climate literature reviews (Stewart, 2011a; Stewart and Oke, 2012). And even though this strategy gives rise to other limitations and challenges (e.g. having only indirect observations available, or bumping into spatial and temporal resolution mismatches), it does however reveal the holistic nature of the LCZ typology that distinguish urban surfaces accounting for their typical combination of micro-scale land-covers and associated physical properties (Demuzere et al., 2020a).

**Table 1.** A selection of urban canopy parameter data associated with built LCZ types, sourced from Stewart and Oke (2012). Columns represent the urban canopy parameters included in the thematic benchmark: the percentage of built ($\lambda_B$ [%], ratio of building plan area to total plan area), impervious ($\lambda_I$ [%], ratio of impervious plan area (paved, rock) to total plan area), and total impervious surface area ($\lambda_T$ [%], defined as the ratio of the sum of the building and impervious plan areas to the total plan area), the mean height of roughness elements $H$ [m] (geometric average of building heights), and the anthropogenic heat flux $AHF$ [W m$^{-2}$]. Maximum values for $H$ (LCZs 1 and 4$^*$) and $AHF$ (LCZ 10$^{**}$) are not available and are arbitrarily set to 200 m and 1000 W m$^{-2}$ respectively.

| LCZ | $\lambda_B$ | $\lambda_I$ | $\lambda_T$ | $H$ | AHF |
|---|---|---|---|---|---|
| 1. Compact high-rise | 40–60 | 40–60 | >80 | >25$^*$ | 50–300 |
| 2. Compact midrise | 40–70 | 30–50 | >70 | 10–25 | <75 |
| 3. Compact low-rise | 40–70 | 20–50 | >60 | 3–10 | <75 |
| 4. Open high-rise | 20–40 | 30–40 | 50–80 | >25$^*$ | <50 |
| 5. Open midrise | 20–40 | 30–50 | 50–90 | 10–25 | <25 |
| 6. Open low-rise | 20–40 | 20–50 | 40–90 | 3–10 | <25 |
| 7. Lightweight low-rise | 60–90 | <20 | >60 | 2–4 | <35 |
| 8. Large low-rise | 30–50 | 40–50 | >70 | 3–10 | <50 |
| 9. Sparsely built | 10–20 | <20 | 10–40 | 3–10 | <10 |
| 10. Heavy industry | 20–30 | 20–40 | 40-70 | 5–15 | >300$^{**}$ |

This approach is in line with previous regional works, that used datasets available for specific regions only, such as for Europe (Demuzere et al., 2019a) or the continental United States (Demuzere et al., 2020a). For the current study, (semi-)independent, consistent and open-source datasets with global coverage are selected, that are critical to distinguish the LCZ classes (surface





cover, packing and height of roughness elements, and thermal properties), and that are ideally representative for the year 2018. The various products are described first, followed by an explanation on how the thematic benchmark is performed.

• **Surface cover** is sourced from the Copernicus Global Land Cover Layers - Collection 3 (CGLCL3), a global discrete land cover map at 100 m resolution, available on a yearly basis from 2015 to 2019, of which 2018 is selected (Buchhorn et al., 2020a, b). These maps describe the Earth's terrestrial surface in up to 23 distinct land cover classes following the United Nations Land Cover Classification System (Di Gregorio, 2005). In contrast to the natural classes, which are primarily obtained via PROBA-V sensor data, the single urban class is largely identified using the World Settlement

Footprint (WSF, Marconcini et al. (2020)) from DLR (German Aerospace Center), a global map of human settlements on Earth for the year 2015.

    • **Packing of the roughness elements** can be characterised by the building ($\lambda_B$), impervious ($\lambda_I$), or total impervious ($\lambda_T = \lambda_B + \lambda_I$) surface. Recent literature reports on a variety of products that claim to represent global impervious surfaces (e.g. Gong et al., 2020; Marconcini et al., 2020; Zhang et al., 2020a). These datasets generally adopt an urban

mask approach; here we follow the European Environmental Agency's (EEA) definition of imperviousness density as 'the percentage of sealed artificial surface' (European Environment Agency, 2018a). In contrast, the global and high-resolution Sentinel-2 based probability of built-up areas (GHS-S2Net) provides a valuable alternative Corbane et al. (2021). GHS-S2Net is produced using a Convolution Neural Networks architecture for pixel-wise image classification that automatically extracts built-up areas at a spatial resolution of 10 m from a global composite of Sentinel-2 imagery

(Corbane et al., 2020), representative for 2018. The dataset reports about built-up areas in the form of probabilities, indicating the probability of a pixel (values between 0-100) to belong to the built-up class. Moreover, based on an evaluation using building footprints from 277 regions across the globe, Corbane et al. (2021) indicated that there is a strong relationship between the output probabilities and the building densities, suggesting that the model outputs can be used as a proxy for $\lambda_B$. As an additional test, we regress the GHS-S2net built-up probabilities against EEA's

100 m imperviousness density (IMD, reflecting $\lambda_T$) and share of built-up (SBU, reflecting $\lambda_B$) layers for the year 2018 (European Environment Agency, 2018a, b), for the thirty largest European functional urban areas (FUAs, Schiavina et al. (2019), see also Appendix C). The results (described in Appendix D) indicate that the GHS-S2net built-up probabilities on average explain >90% of the observed $\lambda_T$ and $\lambda_B$ variability, with regression slopes closer to 1 for $\lambda_B$. Therefore, GHS-S2net built-up probabilities are used in this study as a proxy for $\lambda_B$.

• **Height of the roughness elements** (building height, $H$) data is taken from the 3D building structure data (unpublished data, based on Li et al. (2020a)), a global 1 km² resolution database of building height, building footprint, and building volume estimated for the nominal year of 2015. The data is estimated using a random forest algorithm, based on a wide range input layers, including optical imagery (different Landsat bands), synthetic-aperture radar data from Sentinel-1, derivatives of remote sensing products (such as the enhanced vegetation and normalized difference vegetation indices),

and other socio-economic data (road networks, DEM, Gross Domestic Product and Gini indices reflecting economic inequalities within cities, etc). In this product, building height denotes the average height of all buildings in a pixel,



weighted by the area of each building. As such, it does not consider ground surfaces (roads, parking places, etc.) and excludes other tall features such as trees. Building height is estimated only for areas (pixels) that include built-up land in the year 2015, according to the WSF data (Marconcini et al., 2020).

• **Anthropogenic heat** ($AHF$) is the final LCZ attribute that can be evaluated; unfortunately there are no global databases of thermal and radiative properties of the urban fabric that can be used. Here, the recent 1 km$^2$ global $AHF$ dataset from Varquez et al. (2021) is selected (hereafter referred to as AH4GUC) to benchmark the global LCZ map. AH4GUC is a freely available database (Varquez et al., 2020), contains maps of hourly and annual mean anthropogenic heat emissions representing the periods 2010s and 2050s (2010's annual mean is used here), and integrates anthropogenic

265       heat emissions from primary energy consumption (e.g. industrial, agricultural, commercial, residential and transport sectors) and metabolic processes.

The thematic benchmark is performed for 150 selected urban regions (Fig. C1), which are identified by selecting the 10 most populated FUAs per urban ecoregion that are covered by the global LCZ map. In order to compare the surface cover (built versus natural) from CGLCL3 with the LCZ map, the latter is converted into a binary product; all built LCZs (except

LCZ 9 - Sparsely built, which is predominantly natural) are converted to a single 'urban' class, and all remaining classes are considered as natural. A per-pixel quality assessment is then performed for each FUA, and is described in terms of the balanced accuracy (BA) - providing information about the rate of correctly classified pixels in an unbalanced setting where natural pixels are predominant compared to urban pixels - and Cohen's Kappa (CK) - that compensates for random chance in the pixels assignment (Corbane et al., 2021).


As can be seen from Table 1, the benchmark UCPs $\lambda_B$, $H$, and $AHF$ are characterised by value ranges (e.g. $\lambda_B$ for LCZ 1 ranges between 40 and 60%), so that a one-to-one evaluation is not possible. As such, for each of the UCPs, the mapped LCZ classes are replaced by their corresponding minimum, mean and maximum UCP values, which are then regressed against the reference products described above. As observed $H$ and $AHF$ products are available on a 1 km$^2$ resolution, the 100 m LCZ-

based minimum, mean and maximum UCP maps and 10 m GHS-S2Net urban probabilities are all resampled to a common 1 km$^2$ resolution. The resulting coefficients of determination (R$^2$) and slopes are reported as measures for the LCZ-based UCP explanatory power.

## 3   Results

### 3.1   Global LCZ map

Applying the TA curation procedure explained in Section 2.1 resulted in 410 ROIs, consisting out of 63.847 polygons and 2.018.916 pixels. Their distribution between ER varies (e.g. number of ROIs ranging between 8 and 100, for ER 11 - Tropical, sub-tropical grassland and ER 8 - Tropical, sub-tropical forest in Asia, respectively), in line with global population density patterns (Fig. 3). The number of TA polygons are well distributed across the different LCZ classes (Fig. E1), with lowest

numbers for LCZs 7 (Lightweight lowrise) and 1 (Compact highrise), and highest numbers for LCZs D (Low plants) and 6
(Open lowrise). It is interesting to note that, for most LCZ classes, the biggest share of TA polygons per LCZ class and urban
ecoregion comes from ROIs in ER 3 - Temperate forest in East Asia (Fig. E1), even though this ER only has an average number
of ROIs. This is in part caused by a small number of LCZ Generator submissions with a very high number of TAs, such as the
3000+ TAs for the larger Nanjing - Bengbu - Huai'an (People's Republic of China) submission (Pan, 2021).

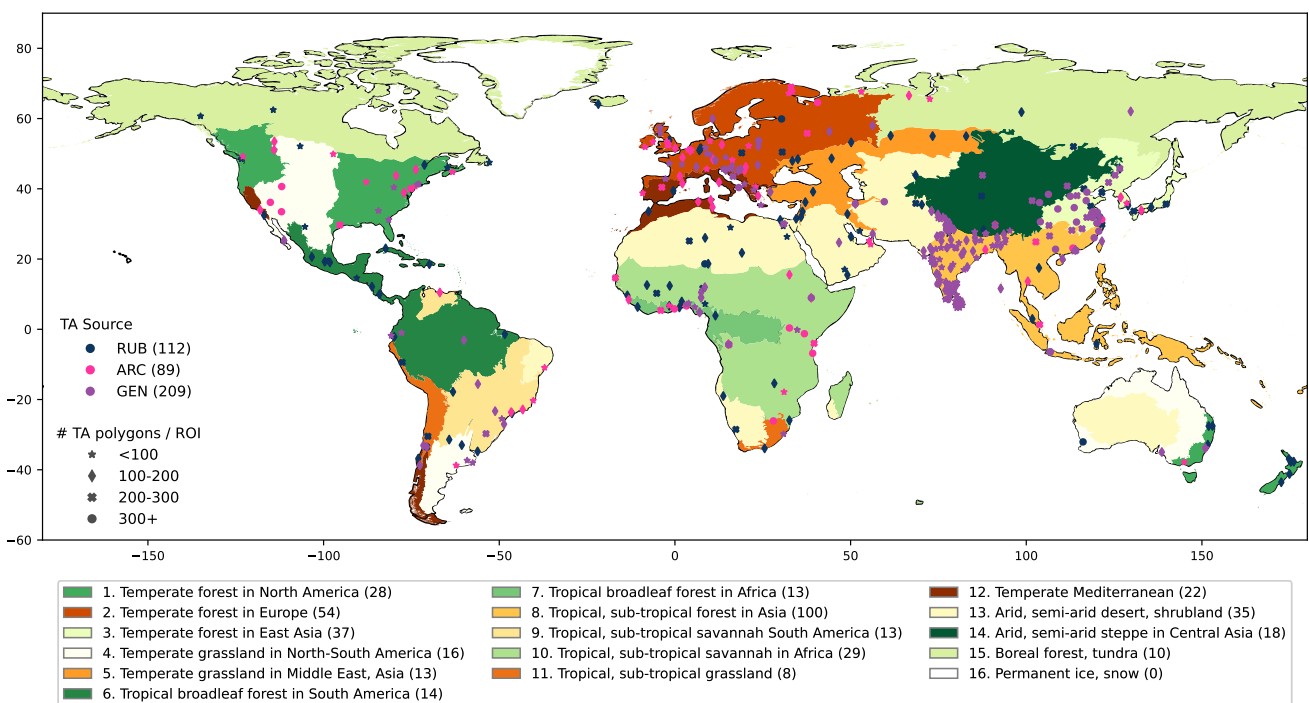

**Figure 3.** Spatial distribution of global TA sets on top of the urban ecoregions (ER), only showing the centroid of each ROI. Marker type
reflects the amount of TA polygons per ROI, marker colors the TA source. Values between brackets in the TA source and ER legend indicate
the number of ROIs per source and per ER, respectively. Note that ER colors and names are adopted from Schneider et al. (2010).

Alongside the curated TA samples, thirty earth observation input features are used in the classification procedure, an outcome
of the feature importance ranking procedure (Section 2.3). The sum average (savg) GLCM texture feature derived from PAL-
SAR's HV polarisation backscattering coefficients is found to be most important, followed by the newly developed NANTLI
metric and the 90th percentile of the Normalized Difference Vegetation Index (NDVI) multi-annual composite (Fig. F1). The
remainder of the selected features contain information about topography, Landsat 8 bands and band ratios, Sentinel-2 NDVI
band ratios, Sentinel-1 VV and VH composites, some other PALSAR HH and HV GLCM textures, and the Global Canopy
Forest Height (GCFH). For clarity, a final list of selected features and their description is provided in the Table F1. Finally, it
is worthwhile to note that the 16 discarded features only indicate a very limited contribution to the LCZ map quality across
all urban ecoregions. For example, six features never belong to the top 50% of the most important features (VVH, PAL-



SAR_HH_SHADE, ASPECT, S2_sei_median, S2_csi_median, and CHM), and another five only one time (S2_B6_median, S2_B7_median, S2_rep_median, VV_HH and EBBI) (please refer to Table 2 in Demuzere et al. (2021b) for abbreviations).

This indicates the generic character of the selected earth observation input feature space that is able to cover the global (urban) land surface heterogeneity representative for different clusters of climate, vegetation, and urban topology.

The resulting 100 m spatial resolution global LCZ classification, based on all TAs and selected input features, is shown in Fig. 4. As LCZs were originally designed as a new framework for UHI studies Stewart and Oke (2012), they also contain a limited

set of 'natural' land-cover classes (LCZs A to G) that can be used as 'control' or 'natural reference' areas, which dominate the global view. However, the seven natural classes in the LCZ scheme can not capture the heterogeneity of the world's existing natural ecosystems, and thus cannot match other products such as the 20, 36 or 75 layers that describe the Earth's terrestrial surface in the Copernicus Global Land-Cover Layers (Buchhorn et al., 2020a, b), the European Space Agency Climate Change Initiative land-cover map (ESA, 2017), or the global map of terrestrial habitat types (Jung et al., 2020) respectively. In contrast,

the added value of the LCZ framework (and map) is the diversity of urban classes, which are easily interpretable and globally consistent, capturing the intra-urban variability of surface forms and land functions.

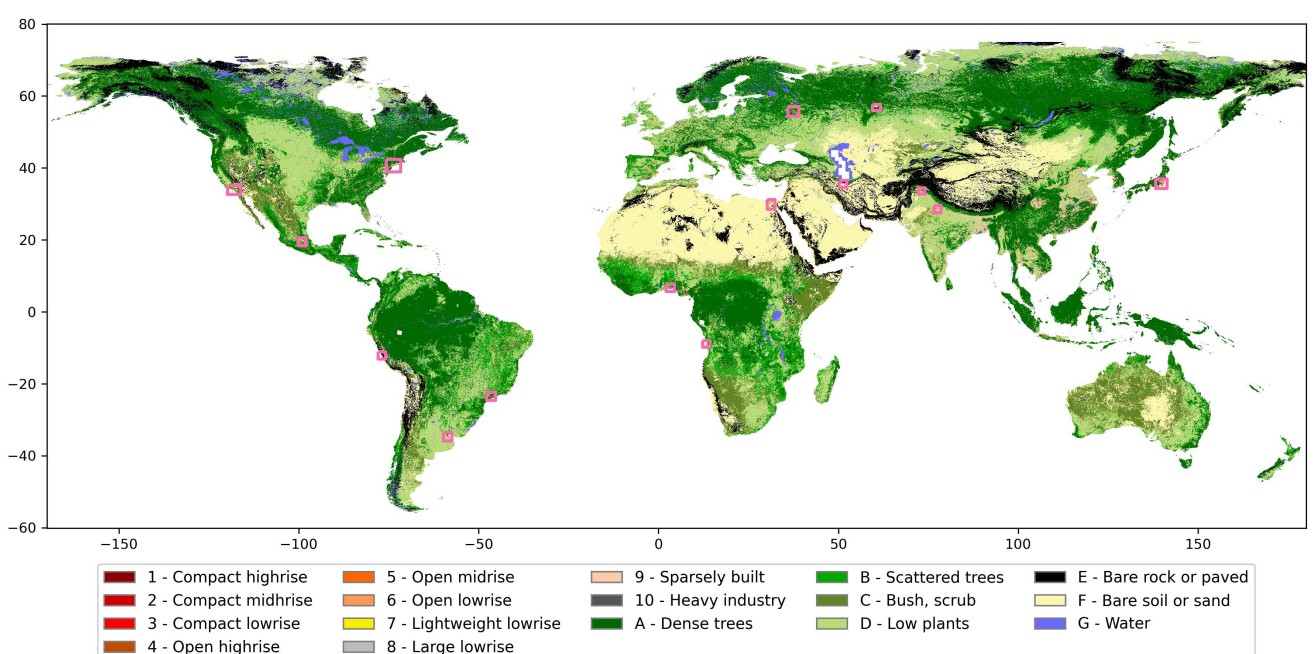

**Figure 4.** Global map of Local Climate Zones. Detailed views of the pink bounding boxes are shown in Figures 5, 6 and 7.

This is show-cased by zooming into the largest FUAs per urban ecoregion in Figures 5, 6 and 7, simultaneously showing the LCZ probabilities (discussed in Section 3.2) and corresponding binary CGLCL3 surface cover. The LCZ map for e.g. New York (ER1, Fig. 5) illustrates the compact high- and mid-rise areas clustered in and around Manhattan, more open and lower-





rise areas outwards of the city, and large-scale low-rise and industrial urban urban land cover around the Port of Newark west
of Manhattan. A second example is the city of Moscow (ER2, Fig. 5) in which its concentric layout mainly hosts LCZs 1, 2
and 4 in the centre, and LCZs 5 and 6 when moving to the suburbs and its satellite cities. Such information is crucial to e.g.
characterise the UHI, as was recently demonstrated for Moscow by Varentsov et al. (2020, 2021), using the regional climate
model COSMO-CLM, and observations from a dense network of around 80 reference stations augmented with more than 1.500

crowd-sourced citizen weather stations. More in general, the global LCZ map allows to make such type of assessments for any
global urban area, by moving away from the traditional urban mask and incorporating cities' internal make-up (Bechtel et al.,
2017; Ching et al., 2018).

**Figure 5.** LCZ map, its probability and the corresponding binary CGLCL3 surface cover for the largest FUAs in urban ecoregions 1 to 5.



Open Access Earth System Discussions
Science
Data



**Figure 6.** As Fig. 5 but for the largest FUAs in urban ecoregions 6 to 10.





**Figure 7.** As Fig. 5 but for the largest FUAs in urban ecoregions 11 to 15.





Historic urbanisation patterns are the consequence of countless decisions made at building, neighbourhood and city scales. As such, cities have unique fingerprints reflecting distinct topographic, cultural and economic contexts. To assess the global

differences of built forms and functions, the LCZ frequencies are first categorised in groups reflecting their degree of total impervious fraction. The HIGH-$\lambda_T$ cluster (LCZs 1, 2, 3 and 8) is characterised by average $\lambda_T > 85\%$, whilst the MEDIUM-$\lambda_T$ cluster (LCZs 4, 5, 6 and 9) typically has average $\lambda_T$ values between 25% and 70%. A third cluster is added that groups LCZs 7 and 10, two LCZ classes that are distinct for their materials (LCZ 7) and anthropogenic heating (LCZ 10). The distribution of these clusters is then aggregated and visualised per urban ecoregion, enriched by their corresponding underlying LCZ classes

and their lcz-based building height properties (Fig. 8). It is clear that there are fundamental geographic differences in the urban layout of cities: cities in e.g. ER 1 (Temperate forest in North America) are dominated by the open cluster, and more specifically LCZ 6. The small fraction taken by the compact cluster is in turn dominated by LCZ 8, reflecting large lowrise buildings - often commercial centers. In contrast, cities in ER 11 (Tropical, sub-tropical grassland) have a more balanced distribution of compact and open classes. Here, the compact class is mostly occupied by LCZs 3 and 8, and the open cluster by LCZs 6 and 9. One

consistent pattern is however apparent: a clear domination of low-rise built forms across all urban ecoregions. The two most contrasting examples in this respect are ER 3, with a relevant share of LCZs 1, 2, 4 and 5, and ER 10 (Tropical, sub-tropical savannah in Africa) that is almost completely dominated by low-rise built forms.

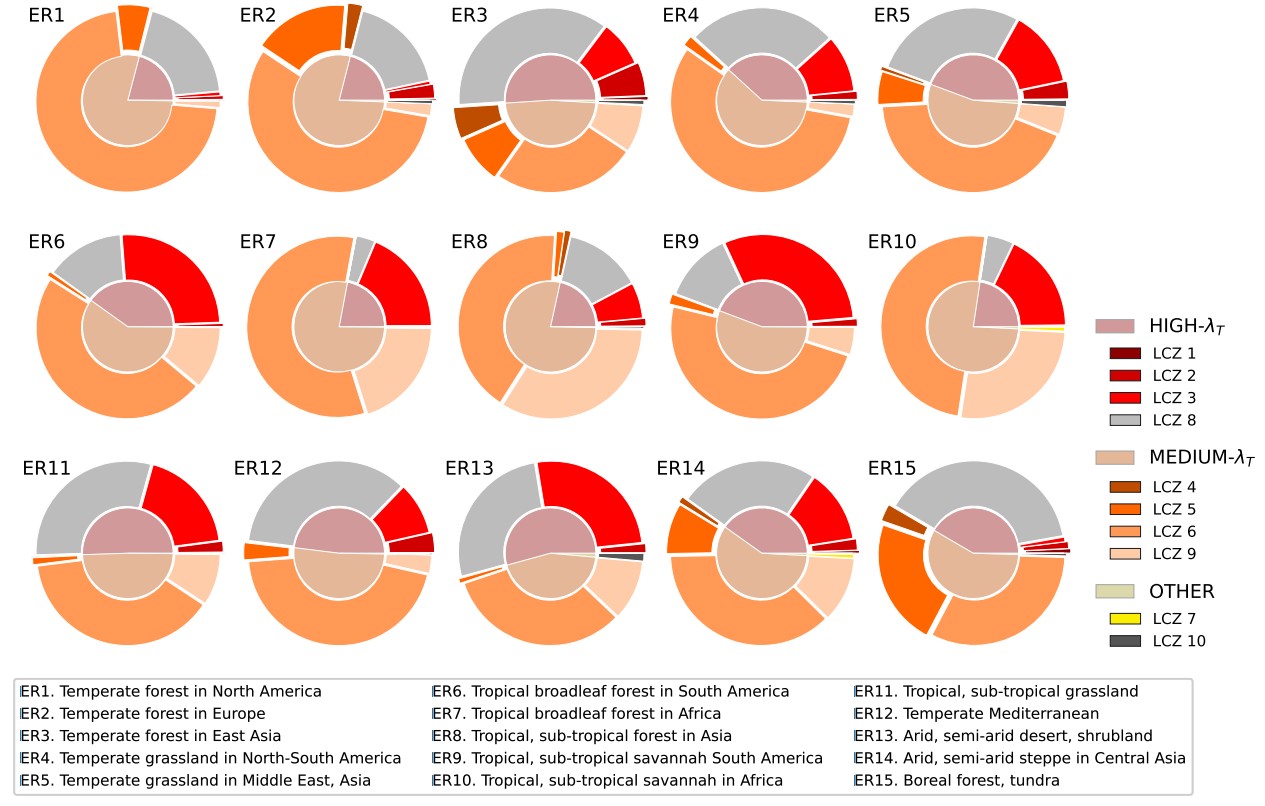

**Figure 8.** Distribution of the built LCZ classes for all 13.135 urban centres in the Urban Centre Database, aggregated per urban ecoregion. The inner rings indicate the HIGH (LCZs 1, 2, 3, and 8), MEDIUM (LCZs 4, 5, 6, and 9) and OTHER (LCZs 7 and 10) degree of total imperviousness ($\lambda_T$) clusters. The outer rings depict the actual LCZ classes. The expansion of individual LCZ wedges visually reflects the differences in building height across LCZ classes (see Table 1).

## 3.2 Quality assessment and benchmarking

As part of the multiple lightweight global random forest models procedure described in Section 2.3, a global LCZ-probability layer is produced that identifies how often the modal LCZ was mapped (out of a total of 50 random forest results), indicating a first measure of robustness of the classification. This probability layer (in %) is shown in the middle panels of Fig. 5 for the largest FUAs per urban ecoregion. Yet in order to get a more comprehensive overview, LCZ-based probabilities are aggregated over all 13.135 cities in the Global Human Settlement Layer Urban Centres Database (GHS-UCDB, Florczyk et al. (2019)). Mean probabilities across the globe are greater than 50% for all LCZ classes, meaning that the resulting modal LCZ class was mapped by more than half of the 50 LCZ models. Highest probability values are obtained for LCZs 6, 8, A (Dense trees) and G (Water) (∼80 to 100 %), and lowest values are found for LCZs 1, 4, 5 and 7, which can be due to a variety of reasons. First, these LCZ types are typically characterised by a lower number of TAs, decreasing its potential weight in the random forest



models. Second, some of these LCZ classes are characterised by similar building footprints and impervious surface areas
(e.g. LCZs 4 and 5), yet differ mainly in the height of their roughness elements (see Table 1). As a conscious decision was
made to use the lower resolution building height dataset as a semi-independent benchmark dataset (described in Section 2.4.2),
currently no input feature directly represents the roughness of buildings (See also Demuzere et al., 2019b, 2020a). In terms
of the ER-stratified values, all probabilities per LCZ class are in line with the global values, demonstrating the universality
of the LCZ typology and the robustness of the classifiers and input features across the urban ecoregions. Probabilities for
LCZ 7 deviate from this behaviour (values < 40% for ER's 11 and 12), which might be due to relatively low number of ROIs
and concurring lower number of TA polygons for LCZ 7. In addition, as this LCZ class includes informal settlements - often
consisting of lightweight materials and densely packed buildings inter-spaced with hard-packed surfaces - these pixels present a
challenge for the classifier because of their mixed spectral signature (Stewart, 2011b; Brousse et al., 2020a; Van de Walle et al.,
2021, 2022). For this LCZ type, future versions of the global map might benefit and built further upon recent efforts dedicated
to map informal urban settlements (see e.g. Kuffer et al., 2020; Assarkhaniki et al., 2021; Owusu et al., 2021; Abascal et al.,
365 2022).

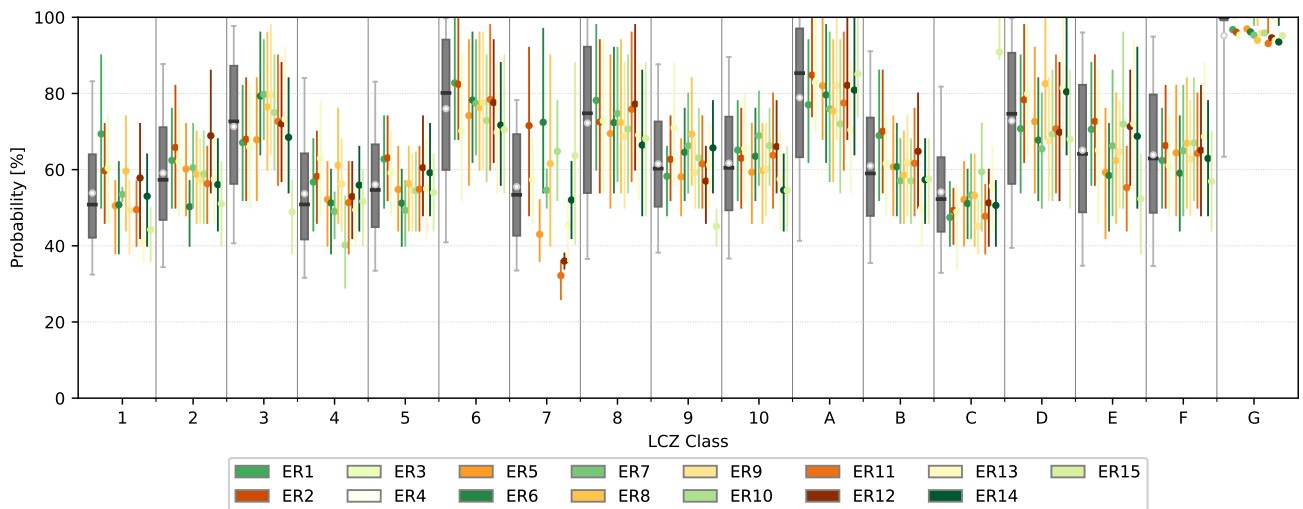

**Figure 9.** Probabilities of the mapped LCZ classes, aggregated over all GHS-UCDBs. The grey boxplots depict the probability distribution
for all global urban centres, per LCZ class, with boxes and whiskers spanning the 25-75 and 5-95 percentiles respectively, and means and
medians indicated by the white dots and black lines respectively. The ER-colored lines (color legend as in Fig. 3) indicate the 25 to 75th
percentile range averaged over the urban centres, stratified per ER. ER-colored dots indicate the mean.

The traditional accuracy assessment using the independent training / test samples obtained via the 'splitting the polygon
pool' approach (Xu et al., 2021) and the fifty lightweight random forest models results in scores of >70% for all OA metrics
(Fig. G1). The variability across the fifty random forest models is small, indicating the robustness of the global classification
protocol. Interestingly, the global OA values using the reduced set of final input features is higher compared to the accuracy





assessment using all input features (74.5% ± 15.1, Fig. F1), supporting the valid removal of uninformative input features from the multi-dimensional input feature space. The class-wise F1 metric shows larger variability with values for the built LCZs between 50% (LCZ 1 - Compact highrise) and 78% (LCZ 6 - Open lowrise), and >60% for all natural classes. The lowest accuracy is obtained for LCZs 1, 4, 5, 7 and 10, in line with the results of the probability layer discussed above.

Since the LCZ typology is a representation of urban form - defined via the corresponding universal LCZ-based canopy parameters - a thematic benchmark allows to indirectly assess the quality of the LCZ map for continuous land surfaces, including those pixels not part of the TA samples used in the traditional accuracy assessment. Fig. 10(A) reveals a good correspondence between the built LCZ classes and the urban class from the CGLCL3 - taken from the World Settlement Footprint data (Marconcini et al., 2020) - with an average Balanced Accuracy (BA) and Cohen's Kappa (CK) of 90% and 73%, respectively.

Stratifying the BA results per urban ecoregion indicates a similar performance, with mean BA values ranging between 83% (ER3) and 93 % (ER4) (Fig. H1). A similar variability can be observed for the CK results stratified per urban ecoregion, with mean CK values ranging between 65% (ER8) and 80% (ER9). These results indicate that the global LCZ map presents a good correspondence with a state-of-the-art and dedicated built-up land data product, and is thus able to correctly discriminate between built-up and natural land cover (confirmed independently by an $OA_{bu}$ of ~95%, an accuracy metric that evaluates the

built versus natural LCZ classes only (Fig. G1)).

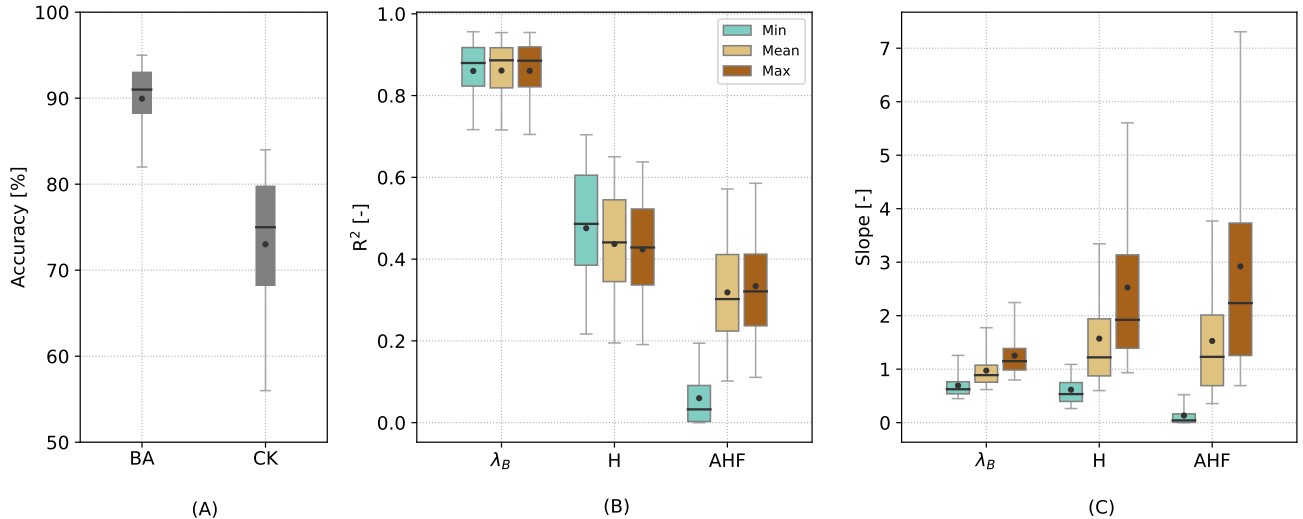

**Figure 10.** Results for the thematic benchmark, for (A) the CGLCL3 urban mask (A) and $\lambda_B$, $H$, and $AHF$ (B and C). All accuracies are derived for the 150 global LCZ FUA regions. For built-up land, accuracy is expressed using the Balanced accuracy (BA) and Cohen's Kappa (CK). For $\lambda_B$, $H$, and $AHF$, the coefficients of determination ($R^2$) (B) and slopes (C) result from the regression between the reference datasets and their corresponding universal LCZ-based values from Stewart and Oke (2012), using minimum, mean and maximum values (colours). For all panels, boxes and whiskers span the 25–75 and 5–95 percentiles respectively. The means and medians are indicated by the black dots and lines respectively. Results stratified per urban ecoregion are available in Appendix H.





For $\lambda_B$, $H$, and $AHF$ - characterized by UCP value ranges (Table 1) - a one-to-one evaluation with their reference datasets (described in Section 2.4.2) is not possible. As such, minimum, mean and maximum UCP values are regressed against these reference products, from which the coefficients of determination ($R^2$) and slopes represent the measures of explanatory power of the LCZ map (Fig. 10B,C and Fig. H2 for results stratified per urban ecoregion). The LCZ-based building surface fraction

$\lambda_B$ is in very good agreement with the GHS-S2Net proxies, with mean $R^2$ values close to 0.9 for the whole value range. Slope values provide a slightly more nuanced result, with average values 0.69, 1 and 1.22 when targeting the minimum, mean and maximum $\lambda_B$ UCP values. In other words, assigning the means of the $\lambda_B$ value ranges to each corresponding LCZ class is a very good approximation for the building surface fraction for global cities. Results for building heights $H$ and the anthropogenic heat flux $AHF$ can be interpreted in the same way (Fig. 10B,C and Figures H3 and H4 for results stratified per

urban ecoregion). For $H$, $R^2$ / slope values range between 0.42 / 2.5 and 0.5 / 0.59 for the maximum and minimum LCZ-based $H$ values. Also here, the best results are obtained using the mean of the LCZ-based value ranges, even though only ∼50% of the observed building height is explained, and the mean LCZ-based values tend to overestimate the reference values. For $AHF$, it is clear that assigning the minimum of the LCZ-based $AHF$ range to the LCZ map has little explanatory power: $R^2$ is 0.1 with a slope of ∼0.1. Sightly better results are obtained when using the maximum and especially the mean of the $AHF$ value range,

but also here only 30% of the observed $AHF$ variability can be explained by the LCZ map. From previous anthropogenic heat flux works (Oke et al., 2017) it is clear that this is not surprising. The $AHF$ values provided by Stewart and Oke (2012) are fixed ranges, reflecting the mean annual heat flux density from fuel combustion and human activity (transportation, space cooling/heating, industrial processing, human metabolism) at the local scale. Yet as also indicated in their Table 4 (footnote c), these values vary significantly with latitude, season, and population density. LCZs have previously been shown to indirectly

capture information on population densities (Demuzere et al., 2020a), and also the role of seasonality with respect to LCZ-based annual $AHF$ values has been discussed elsewhere (e.g. Varentsov et al. (2020)). Yet it is obvious that the observed zonal $AHF$ variability in AH4GUC (Varquez et al., 2021, their Fig. 5) is neglected completely in this thematic benchmark. This means that, for example, an LCZ 6 neighbourhood in tropical Singapore will be assigned the same mean annual $AHF$ values as an LCZ 6 area in the high-latitude city of Helsinki (Finland), even though it is clear that their building cooling /

heating patterns will be completely different (Quah and Roth, 2012; Karsisto et al., 2016). This reveals a strong limitation of using $AHF$ as an independent benchmark of the global LCZ map. Unfortunately, as there are currently no globally explicit databases on thermal and radiative properties of the urban fabric, $AHF$ is currently the only available proxy to indirectly assess the 'thermal' signature of a built environment. Finally, there are a number of other elements that might affect deviations between the LCZ-based UCP parameters and their benchmark products and comparison methods: some benchmark products

are merely indirect observations (e.g. GHS-S2Net urban probabilities being used as a measure for $\lambda_B$), UCPs might have dissimilar definitions (e.g. geometric average of buildings heights versus average of building heights weighted by building footprint), or data matching differences as a consequence of different resolutions and map projections leading to potential artefacts from resampling.



## 4 Serving earth system modelling and urban scale environmental science

Despite the new focus on cities as a critical scale for climate change risk management, we know very little about most cities on the planet - being generally ignorant of their extent, how they are constructed and how they are occupied (Demuzere et al., 2020a). This knowledge gap is especially true for urban areas in the Global South, where 90% of the projected world population growth of 2.5 billion over the next couple of decades will occur. This is in strong contrast with current urban knowledge that is predominantly shaped by research on and from the Global North (Nagendra et al., 2018). The global Local Climate Zone map

presented here provides a globally consistent and climate-relevant urban description, that is an important prerequisite for developing fit-for-purpose integrated climate-sensitive urban planning policies (Georgescu et al., 2015). As LCZs are developed from generalised perspectives of built forms and land cover types that are universally recognised and applicable (Stewart and Oke, 2012), this global LCZ map provides standardised and harmonised data of all cities, allowing to consistently assess the heterogeneous nature of cities' urban forms and functions, and providing the much-needed platform for comparative analyses,

systematic learning and horizontal knowledge exchange between cities and regions (Raven et al., 2018; Ching et al., 2018; Bai et al., 2018; Creutzig et al., 2019; Reba and Seto, 2020).

    As cities are complex systems and their components are difficult to understand in isolation, fundamental scaling laws that seem universal across cities are often put forward to understand their dynamics, growth and evolution in a scientifically pre-

dictable, quantitative way (Bettencourt et al., 2007, 2020). In this work, city growth, structures and functions (e.g. urban area, albedo, population density, building density, building heights, anthropogenic heat flux, and sky view factor) are completely represented by population and universal scaling laws (Schläpfer et al., 2015; Manoli et al., 2019; Martilli et al., 2020). Other work has explored the correspondence between population density and carbon dioxide emissions (Ribeiro et al., 2019) and the allometric-scaling relationships between settlement population and non-point-source emissions of air pollutants (MacKenzie

et al., 2019). In reality, the global universality of these scaling laws is unknown as many are created based on regional information only, mostly for data rich parts of the world (Bettencourt and West, 2010). Currently, city population is often used as proxy for urban form but the global LCZ map offers a richer alternative. In addition, the combination of population and LCZ may provide deeper insights into the variations of form in different cultural, socioeconomic and climatic contexts, and help guide future urban development. Since LCZs distinguish urban areas based on their form, the global map provides the means

to assess the universality of the above-mentioned scaling laws, and refine/improve them. In addition, the global LCZ map has value beyond climate applications when combined with other spatially resolved urban information (Reba and Seto, 2020), like flooding hazard, biodiversity and air quality, for example.

    Since the LCZ typology was initially designed for urban temperature studies (Stewart and Oke, 2012), typical applications

focus on the UHI, usually providing the context for designing and analysing observations from urban meteorological networks (Skarbit et al., 2017; Beck et al., 2018; Chieppa et al., 2018; Verdonck et al., 2018; Yang et al., 2018; Leconte et al., 2020; Milošević et al., 2021; Zhang et al., 2020b; Zong et al., 2021), from crowdsourced data (Fenner et al., 2017; Varentsov et al.,





2021; Fenner et al., 2021) or from remote sensing (Wang and Ouyang, 2017; Bechtel et al., 2019b; Eldesoky et al., 2021; Stewart et al., 2021). However, the typology has been used for other purposes (see also Lehnert et al. (2021) for European

applications), such as urban heat (risk) assessment studies (Verdonck et al., 2019b; Van de Walle et al., 2022), climate sensitive design, land use/land cover change, and urban planning (policies) (Perera and Emmanuel, 2018; Aminipouri et al., 2019; Vandamme et al., 2019; Maharoof et al., 2020; Chen et al., 2021b; Cai et al.), anthropogenic heat, building energy demand and consumption, and carbon emissions (Wu et al., 2018; Santos et al., 2020; Yang et al., 2020; Benjamin et al., 2021; Kotharkar et al., 2022), quality of life (Sapena et al., 2021), urban ventilation (Zhao et al., 2020b), air quality (Steeneveld et al., 2018; Lu

et al., 2021a), urban vegetation phenology and ecosystem patterns, functions and dynamics (Kabano et al., 2021; Zhao et al., 2022), and epidemiological studies (Brousse et al., 2019, 2020a).

The LCZ scheme is a core element of the WUDAPT project to provide consistent urban data to support climate science (Ching et al., 2018, 2019) and many modelling systems nowadays ingest the LCZ typology, such as e.g. the Surface Urban

Energy and Water Balance Scheme (SUEWS, Alexander et al. (2015, 2016)), UrbClim (Verdonck et al., 2018, 2019b; Sharma et al., 2019; Gilabert et al., 2020), the Vertical City Weather Generator (Moradi et al., 2022), ENVI-met (Middel et al., 2014; Lyu et al., 2019; Bande et al., 2020), the urban multi-scale environmental predictor (UMEP, Lindberg et al. (2018)), MUKLIMO_3 (Bokwa et al., 2019; Matsaba et al., 2020; Gál et al., 2021), COSMO-CLM and the WUDAPT-TO-COSMO tool (Wouters et al., 2016; Brousse et al., 2019, 2020b; Varentsov et al., 2020; Van de Walle et al., 2021), and the Weather Research

and Forecasting model (WRF, Brousse et al. (2016); Hammerberg et al. (2018); Molnár et al. (2019); Pellegatti Franco et al. (2019); Wong et al. (2019); Mu et al. (2020); Zonato et al. (2020); Patel et al. (2020); Demuzere et al. (2021a); Hirsch et al. (2021); Patel et al. (2022)). Most studies focus on individual cities, with the work of Patel et al. (2022) being an exception as it uses the European LCZ map ((Demuzere et al., 2019a)) to simulate a continental-scale heat wave event. The LCZ map presented here allows the extraction of urban data suited to the scale of study and can support global climate and earth system modellings.


Regional climate models are expected to remain indispensable tools that complement global models for understanding physical processes governing regional climate variability and change (Gutowski et al., 2020). Yet at the same time, regional climate model developments also serve as a precursor for the evolution of global climate models, and major efforts are currently underway to increase ESMs to kilometre-scale resolutions (Schär et al., 2020; Bauer et al., 2021). Recently, Fuhrer et al. (2018)

performed a near-global climate simulation at a horizontal grid spacing of just 930 m. Such advancements will represent a quantum jump in (urban) global climate modelling, enabling the explicit treatment of the complex interactions between the fine-grained urban heterogeneity and its atmosphere (Martilli et al., 2020). To date however, climate projections focused on built landscapes are absent, partly owing to the lack of climate-relevant urban data for ESMs (Zhao et al., 2021; Hertwig et al., 2021). Only one ESM included details of urban form in CMIP5[3] (Zhao et al., 2021), and a few more in CMIP6: except for

GFDL-ESM (Dunne et al., 2020), these CMIP6 ESMs all use the Community Land Model-Urban (CLMU) urban canopy parameterisation (Oleson and Feddema, 2020). Yet despite its pole position, CLMU's lead developers indicate that transitioning

---

[3]Coupled Model Intercomparison Project, led by the World Climate Research Programme (https://www.wcrp-climate.org/wgcm-cmip)



to the LCZ urban classes and their corresponding UCPs will likely be beneficial for better simulating the interactions between the urban fabric and the climate system (Oleson and Feddema, 2020). Eventually, a more close connection between the global LCZ map and the ESM community might have a direct impact on climate change policies, via IPCC's[4] upcoming 7th cycle of

Assessment Reports and its planned Special Reports on Cities and Climate Change.

Even though the LCZ typology presents a leap forward in describing intra-urban heterogeneity in a universal manner, its generalisation of course also has its limitations. In the words of Stewart and Oke (2012): "it's view of the landscape universe is highly reductionist [...] and LCZs represent a simple composition of buildings, roads, plants, soils, rock, and water, each in

varying amounts and each arranged uniformly into 17 recognisable patterns. The 17 patterns should nevertheless be familiar to users in most cities, and should be adaptable to the local character of most sites". This multi-urban class typology follows the discourse of most categorical mapping efforts discussed elsewhere (Coops and Wulder, 2019), such that individual LCZ classes are each physically discrete in surface structure and land cover, leading to well-defined boundaries separating most classes. However, users of LCZs must always accept that the internal homogeneity portrayed by each class is unlikely to be found

in the real world, but that the attempt to classify surface complexity in cities represents a key advancement in urban climate science (Stewart and Oke, 2012). It also represents a helpful starting point for more detailed studies of urban form and function at smaller spatial scales. Likewise, due to its reductionist character, the landscape universe represented by the 17 LCZ classes is not complete for several reasons. First we do not apply the subclass scheme, which allows mixture of several LCZ types but reduces its universality. Secondly, some landscapes such as extensive greenhouses are not included in the scheme, since they

are unlikely to be selected for urban heat island studies. Moreover, the scale of real urban structures does not always match the climatic definition of local scale. However, we are convinced that the scheme is the best compromise between climatic variation and generic description of urban structures.

This first version of the global LCZ map itself also has limitations. For example, LCZ 7 requires more attention and alter-

native mapping strategies, as discussed in Section 3.2. Confusion may also exist between classes with similar surface fractions (e.g. LCZs 3 and 8), especially if built materials have similar spectral properties (see e.g. LCZ map for Lima (Peru, ER11), Fig 7). Similarly, also the confusion between classes with similar surface fractions yet different height of roughness elements might be improved in the future. Note that the LCZ map also inherits shortcomings of the many global earth observation input features upon which it is built, such as for example, some missing data in areas that are frequently covered in clouds, or gaps

in coverage because of changing satellite duty cycles. Such limitations can be addressed in future releases of the map, e.g. by harvesting the growing number of TA samples submitted to the LCZ Generator (which received more than 1.500 submissions in less than one year of operation), ingesting more and new high-resolution (earth observation) datasets when available, or by implementing alternative scalable classification algorithms (e.g. Yokoya et al., 2018; Yoo et al., 2020; Rosentreter et al., 2020). Nevertheless, from many above-mentioned examples and applications it is clear that this global LCZ map has a lot of

potential serving urban and climate sciences at various scales. The map is universal and allows for comparisons between global

---

[4]The Intergovernmental Panel on Climate Change



regions. Yet at the same time it is flexible enough to allow anyone to adapt it to suit their purpose, using for example user- and site-specific LCZ-based UCP value if available (Ching et al., 2018). In other words, the global LCZ map describes all the cities of the world in the same, universal language, but interested users can read it in their own dialect. Finally, this development will also support future large-scale dynamic LCZ mapping efforts. Such examples to date are rare and focus on targeted cities

(e.g. Vandamme et al., 2019; Wang et al., 2019; Demuzere et al., 2020b; Zhao et al., 2020a; Lu et al., 2021b; Cai et al.), yet they reveal a large potential in terms of characterising the temporal transformations of urban morphologies across and within different cities, identify the main drivers of such changes, and bridge the gap between policy making and urbanisation patterns, required to come up with informed, data-driven and rational urban planning strategies toward sustainable city developments.

## 5   Conclusions

Since their introduction in 2012, Local Climate Zones (LCZs) have become a standard for characterising urban landscapes according to climate-relevant properties of the surface. From that point forward, the number of applications using this universal urban typology has been growing exponentially, revealing the relevance and potential for a wide range of urban sciences. One of the typology's most popular uses is digital mapping, which can generate UCPs at the city scale for input to numerical climate models. However, the lack of available and consistent global data on the form and function of cities has impeded progress in

urban climate sciences so far, limiting applications to cities or regions for which LCZ maps are currently available.

The 100 m resolution global LCZ map presented here is the first of its kind depicting the much needed global intra-urban heterogeneity in an universal language. It allows easy access to Local Climate Zone data for regional and global scale analysis and provides a seamless integration into existing topographic, natural land cover, and other global scale data products. The map

is generated building further upon previous studies, whilst adding new methodological features that balance optimal learning across global climates and urban typologies with accuracy, computational feasibility and efficiency.

Since this global map identifies the relevant data for planning and climate on neighbourhood, city and global scales, its designed to become part of a basic infrastructure to support a host of studies on exposure to environmental hazards, energy

demand, climate adaptation and mitigation solutions and human health, as examples.

## 6   Data availability

The global Local Climate Zone map, representative for the nominal year 2018 and with a spatial resolution of ∼100 m (EPSG:4326), is available from http://doi.org/10.5281/zenodo.6364594 (Demuzere et al., 2022a). The dataset contains various layers stored as separate GeoTIFF files, including: (1) lcz_filter, the recommended global LCZ map after applying the

morphological Gaussian filter described in Demuzere et al. (2020a), (2) lcz, as (1), but presenting the raw LCZ map before applying the morphological Gaussian filter, and (3) the LCZ probability layer (%) that identifies how often the modal



LCZ from (2) was chosen per pixel. The LCZ maps have the default WUDAPT LCZ color scheme embedded (Fig. 1), and all imagery can be processed using (free) GIS software, e.g. QGIS. In addition, a teaser sample is provided to ease accessibility, providing the LCZ map information for the 15 largest functional urban areas stratified by urban ecoregion. These

GeoTIFF files reflect the underlying data used in Figures 5, 6 and 7 of the manuscript. This teaser dataset is available from https://doi.org/10.5281/zenodo.6364705 (Demuzere et al., 2022b).

*Author contributions.* MD designed the research, with feedback from BB and JK. The RUB training area samples were mostly digitised by CM. JK provided technical support with respect to the LCZ Generator database. MD performed all analysis and developed all visualisations. MD wrote the original draft, with contributions from all other authors.

*Competing interests.* The authors declare no competing interest.

*Acknowledgements.* We acknowledge all WUDAPT contributors and community members for providing the training areas via the portal or the LCZ Generator, with a special thanks to Samira Safaee and Dev Niyogi for providing the large sample of Indian cities. We thank USGS and NASA for the free Landsat data, and the Copernicus programme of ESA for the Sentinel data, all acquired and processed via Google's earth engine. We also like to thank all research institutes and universities for the creation and provision of open datasets, such as

the Copernicus Global Land Cover Layers, the Sentinel-2 based probability of built-up areas, and the anthropogenic heat flux data. Finally, we acknowledge support by the Open Access Publication Funds of the Ruhr-Universität Bochum (RUB, Germany).

*Financial support.* This work was conducted in the context of project ENLIGHT, funded by the German Research Foundation (DFG) under grant No. 437467569. JvV is supported by the Netherlands Organisation for Scientific Research, grant no. VI.vivi.198.008, Guiding human settlements towards sustainable urbanisation.





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



# Appendix

**Appendix A: WUDAPT's digitisation guidelines**

In order to guide a user in the digitisation of LCZ training area polygons, a set of digitisation guidelines are provided on the WUDAPT webpage (www.wudapt.org/digitize-training-areas/). This information is split in two parts, discussing: 1) how to digitize a LCZ polygon using Google Earth and the provided .kml template, and 2) good practice guidelines for digitisation (Figure A1).



| Property | Examples |
|---|---|
| **Size**<br><br>Think at the local scale. Individual buildings do not constitute an LCZ. Look for large homogenous areas that are at a minimum 200 m wide at the narrowest point. Use Google Earth's measurement tool to ensure the area is large enough. | |
| **Shape**<br><br>Avoid complex shapes, as this can lead to mixed spectral information. Simple block shapes however will maximize the homogeneity of the spectral information and the number of available satellite pixels available within the shape. | |
| **Homogeneity**<br><br>If you digitize a training area, the surface characteristics should be similar. In case of doubt, better to digitize different training areas that are homogeneous than one area that is too heterogeneous. | |
| **Borders | Distance**<br><br>Try to keep a minimum distance to other LCZs when classifying. If polygons from different classes are too close to each other, the classifier will receive mixed spectral signals which will affect the quality of the classification.<br><br>Similarly, do not digitize your training area too close to other land cover(s). Also, avoid precise digitization along road or river segments, features that are often too narrow. | |
| **Persistence | Seasonality**<br><br>Avoid construction sites, as they are likely to change LCZ type during a short period of time.<br><br>The surface characteristics of some LCZ types may depend on the seasonality. Agricultural areas (LCZ D, crops) are a typical example, where land cover may flip from bare soil to cultivated land throughout the year. Use the Google Earth's time slider to explore seasonality.<br><br>Keep in mind tidal or seasonal waters. Depending on the time of day or season of the year, shorelines or river beds might be dry or contain water. | |
| **Spatial Distribution**<br><br>Distribute training areas over the entire region of interest, as the same LCZs might differ in their appearance and spectral properties for different parts of your region of interest. | |

**Figure A1.** Good practice guidelines for digitising LCZ training area polygons (© Google Earth 2020).



**Appendix B:  GLCM texture and NANTLI input features**

To date, very few studies tested the added value of texture features (derived from the Gray-Level Co-occurrence Matrix, GLCM) in the LCZ classification procedure. Forget et al. (2018) found that combining features computed from both Sentinel-1 VV and VH backscatter polarisations consistently led to better LCZ classification performances in twelve Sub-Saharan African urban areas, even though the introduction of textures computed from different spatial scales did not improve the classification performances. Along the same lines, Hu et al. (2018) found that Sentinel-1 dual-Pol SAR data (including texture features) can

contribute to the classification of transcontinental cities into several LCZ classes. Also Brousse et al. (2020a) successfully used Sentinel-1 GLCM texture features with an 11 by 11 kernel window size to map nine Sub-Saharan African urban areas, in order to better capture the heterogeneities of built up surfaces.

        Since Sentinel-1 backscatter information is already available in the input feature space, and inspired by the findings of Hay

Chung et al. (2021), PALSAR (Phased Array type L-band Synthetic Aperture Radar, Shimada et al. (2014)) backscattering information was added. Yet not the pure HH and HV backscatter information as in Hay Chung et al. (2021), but rather the GLCM texture features derived from them (Haralick et al., 1973; Conners et al., 1984; Chen et al., 2021a), as follows:

    1. a median 2017-2019 PALSAR composite is created for both the HH and HV polarisation backscattering coefficients;

    2. for each polarisation, the eighteen GLCM texture features are derived, with a 2 by 2 and 4 by 4 kernel window, corre-

1075        sponding to a 50 and 100 m spatial resolution, respectively;

    3. only the contrast, dissimilarity, inertia, sum average, and cluster shade texture measures are retained, as the remaining textures indicated little added value in the LCZ mapping (not shown).

        In addition, also the Visible Infrared Imaging Radiometer Suite (VIIRS) Day/Night Band (DNB), provided by Mills et al. (2013), was used as follows:

1. a median 2017-2019 composite is created from the monthly VIIRS Stray Light Corrected Nighttime Day/Night Band Composites (NTL);

    2. this median NTL image is smoothed with a convolution filter using a radius of 300 meter;

    3. afterwards, the smoothed NTL image as normalised (hereafter referred to as $NTL_{norm}$);

    4. NANTLI is calculated, a Normalized Difference Vegetation Index (NDVI) adjusted NTL Index, according to Eq. B1.

$$NANTLI = \frac{1 + (NTL_{norm} - NDVI)}{1 - (NTL_{norm} - NDVI)} \times NTL \qquad (B1)$$



Note that NANTLI is analogous to EANTLI (Zhuo et al., 2015), yet uses the Landsat 8 NDVI input feature available from Demuzere et al. (2021b) instead of EVI (Enhanced Vegetation Index), introduced to mitigate both saturation problems and
blooming effects of VIIRS data (Zhuo et al., 2018; Zhang et al., 2020a).

The added value of these features are evaluated by mapping 45 cities (three cities per urban ecoregion, characterised by large TA samples with a large number of different LCZ classes) into LCZs, following the procedure of the LCZ Generator Demuzere et al. (2021b), but each time using a different set of earth observation input features. This results in four experiments:

– GEN: the default 33 input features available from the LCZ Generator;

– GEN+NANTLI: GEN and NANTLI

– GEN+NANTLI+P2: GEN, NANTLI and the PALSAR texture features derived with kernel size 2 by 2

– GEN+NANTLI+P4: GEN, NANTLI and the PALSAR texture features derived with kernel size 4 by 4.

Results are evaluated in terms of the overall accuracy metrics (OAs) described in Section 2.4. Figure B1(A) displays the
OAs for the GEN experiment for each individual city sorted by urban ecoregion (ER 1 - 15). Figures B1(B), B1(C) and B1(D) indicate the change in OAs for GEN+NANTLI, GEN+NANTLI+P2, and GEN+NANTLI+P4 respectively, compared to GEN. Adding NANTLI increases the average OAs between 0.3 and 2%, with individual city values ranging between -1 and 5.8% (for Constantine - ER12 and Cologne - ER2, respectively). Adding PALSAR texture features further increases the average OAs between 0.8 and 3.2% (GEN+NANTLI+P2, Fig. B1(C)) and 0.9 and 3.8% (GEN+NANTLI+P4, Fig. B1(D)). Here, individual
city values range between -0.3 (Itanagar, ER14) and 7.8% (Rosario, ER4) for GEN+NANTLI+P2 and -0.9 (Havana, ER6) and 13.8% (Rosaria) for GEN+NANTLI+P4. As such, the VIIRS-based NANTLI input feature together with the PALSAR-BASED GLCM texture features, using a 4 by 4 kernel size, are selected as additional earth observation for the LCZ mapping procedure.



**Figure B1.** Overall accuracies for the various input feature experiments, absolute overall accuracies for all cities for GEN (A), and differences in overall accuracies for experiments GEN+NANTLI (B), GEN+NANTLI+P2 (C), and GEN+NANTLI+P4 (D), compared to GEN. Numbers on the top x-axis indicate the average overall accuracy (change) across all cities. ER refers to urban ecoregion.





**Appendix C: Selected functional urban areas for thematic benchmark**

Functional urban areas (FUAs), as defined by the Organisation for Economic Co-operation and Development (OECD) and
the European Union, are sets of contiguous local (administrative) units composed of a city and its surrounding, less densely
populated local units that are part of the city's labour market (commuting zone). As such, these units not only offer the
opportunity to evaluate more densely built city centres, yet also their sparsely built or natural neighbouring landscapes. The 10
most populated FUAs per urban ecoregion used in the thematic benchmark are depicted in Fig. C1.

**Figure C1.** Spatial distribution of the 10 most populated functional urban areas per urban ecoregion. Note that ER colors are adopted from
Schneider et al. (2010).





**Appendix D: GHS-S2Net versus EEA**

Corbane et al. (2021) illustrated that there is a strong relationship between the output probabilities pg GHS-S2Net and observed building densities, suggesting that the model outputs can be used as proxy for impervious surface fractions. As an additional benchmark, we evaluate the GHS-S2Net built-up probabilities for the thirty largest (in surface area) European FUAs (Paris, London, Dortmund, Katowice, Oslo, Madrid, Budapest, Warsaw, Berlin, Lyon, Copenhagen, Milan, Frankfurt am Main, Toulouse, Cologne, Hamburg, Vienna, Helsinki, Leeds, Rotterdam, Prague, Belfast, Liège, Rome, Nantes, Gothenburg, Munich, Krakow, Zurich, and Istanbul) against two products from the European Environmental Agency (EEA):

1. the 100 m Copernicus High Resolution Imperviousness Density (IMD) layer for 2018 (European Environment Agency, 2018a), a thematic product that indicates the total sealing density ($\lambda_T$), ranging from 0-100%,

2. the 100 m Share of Built-up (SBU) layer for 2018 (European Environment Agency, 2018b), an aggregated version of the 10 m Impervious Built-up map that indicates the building surface fraction ($\lambda_B$), ranging from 0-100%

In line with Bechtel et al. (2019a), all data layers are resampled to a common 1 km grid. Afterwards, EEA's IMD ($\lambda_T$) and SBU ($\lambda_B$) land-only pixels are regressed against the GHS-S2Net built-up probabilities for all thirty European FUAs (see Figure D1 as an illustration), and the corresponding coefficients of determination ($R^2$) and slopes are reported for $\lambda_T$ and $\lambda_B$ separately (Figure D2). Most $R^2$ values exceed 0.9 for both $\lambda_T$ and $\lambda_B$, confirming that GHS-S2net is able to explain most of the observed EEA impervious surface fractions. On average, the slopes of the regression between the built-up probabilities of GHS-S2Net and EEA's IMD and SBU products are 0.6 and 0.7 respectively. These results confirm the findings of Corbane et al. (2021) that the GHS-S2net built-up probabilities can serve as a proxy for $\lambda_B$.



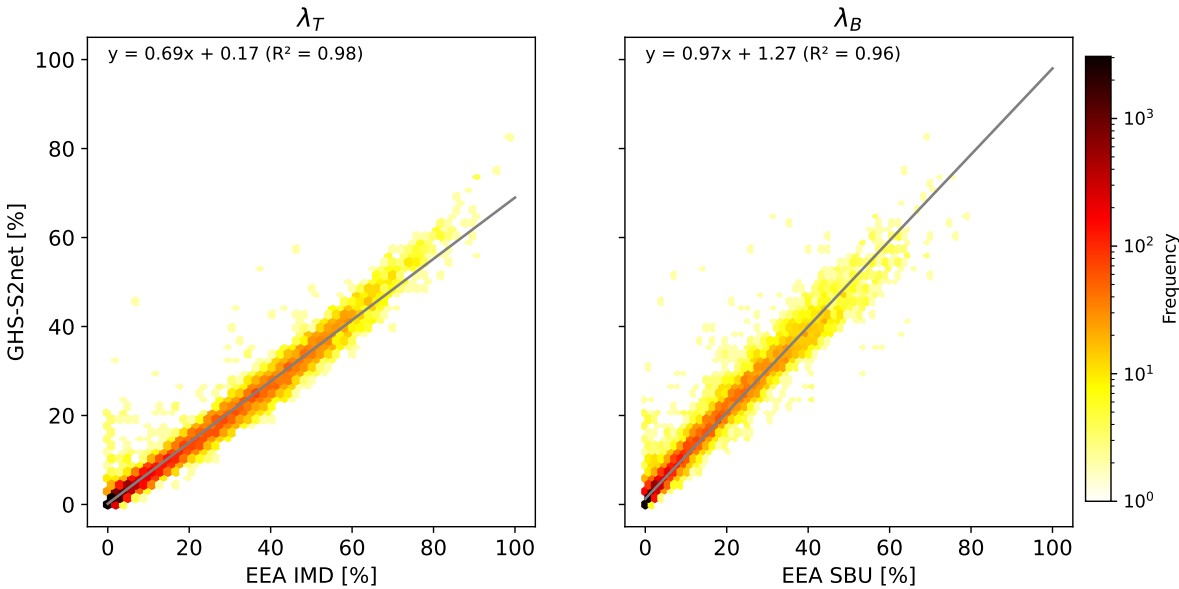

**Figure D1.** Hexbin illustration of the EEA Imperviousness Density (IMD, reflecting $\lambda_T$) and Share of Built-up (SBU, reflecting $\lambda_B$) against GHS-S2Net built-up probabilities for the European Functional Urban Area (FUA) of Dortmund (Germany). Linear regression equations and $R^2$ values are provided for both $\lambda_T$ (left panel) and $\lambda_B$ (right panel). The logarithmic colorbar represents the number of pixels in each imperviousness bin.

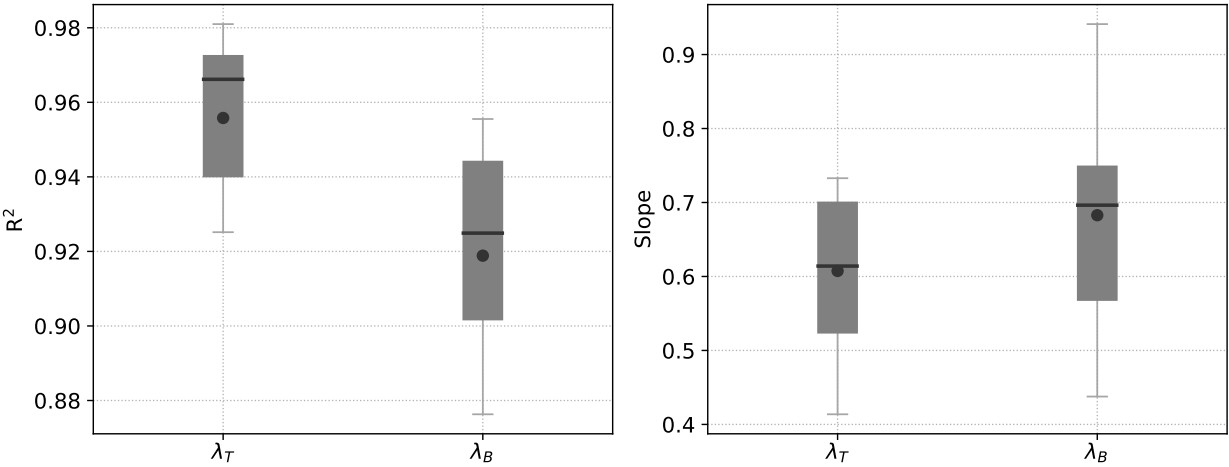

**Figure D2.** Distribution of $R^2$ values and slopes for the regressions between between GHS-S2Net and EEA's IMD ($\lambda_T$) and SBU ($\lambda_B$) datasets, for the thirty selected European FUA boundaries. Grey boxes and whiskers span the 25–75 and 5–95 percentiles respectively. The means and medians are indicated by the black dots and lines respectively.



**Appendix E:  Final number training area polygons per LCZ class**

**Figure E1.** Number of training area polygons per LCZ class and ER (colors as in Fig. 3).





## Appendix F: Final input features

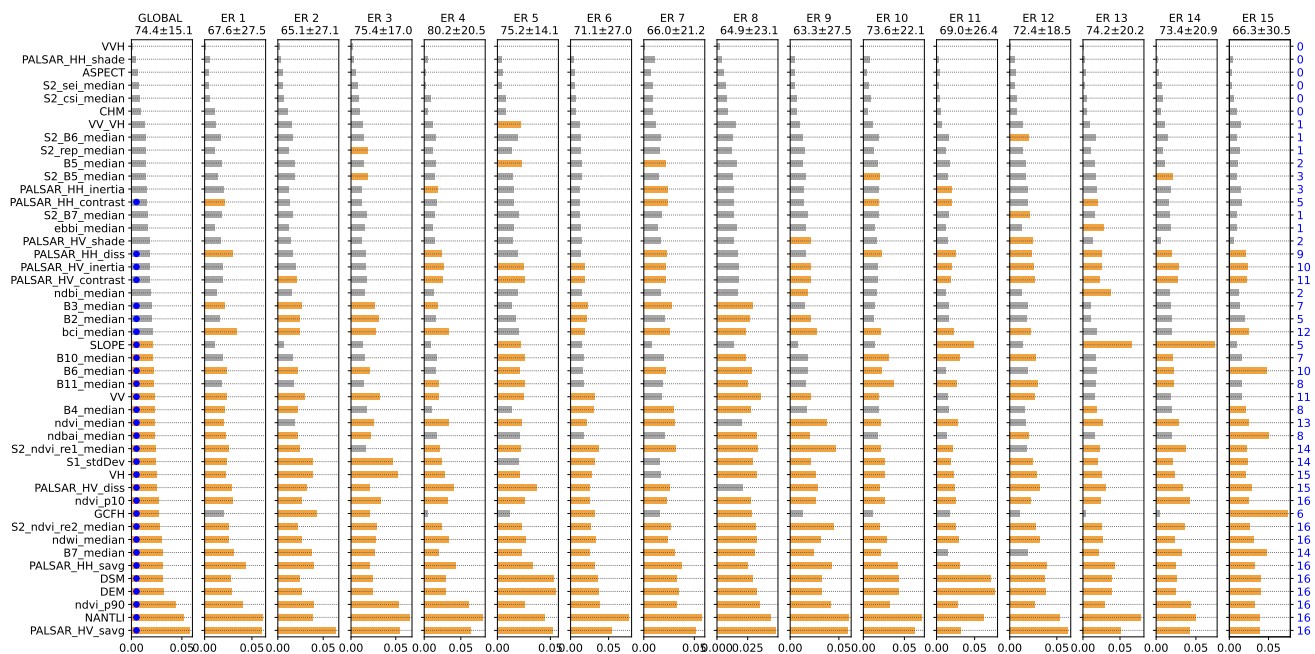

**Figure F1.** Feature importance ranking for the globe and per urban ecoregion (ER). Bars in bright orange depict the input features (per spatial unit) that belong to the top 50% of most important variables. Numbers on top depict the overall accuracy $\pm$ the standard deviation (%). Blue numbers on the right-hand side describe how often an input feature belongs to the top half of most important features. Features with a value $\geq 5$ are used in pathway 2 to create the global LCZ map and are indicated by the blue dot in the global panel on the left. These features are described in more the Table F1.

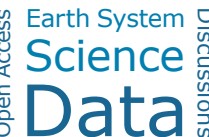

**Table F1.** Final set of input features used in the global LCZ mapping process. The information is structured by sensor, with the input feature names referring to the abbreviations used in the feature importance ranking shown in Fig. F1.

| Sensor | Input feature | Description |
|---|---|---|
| PALSAR | HH_contrast | Contrast GCLM texture parameter from the HH polarisation backscatter |
| | HH_dist | Dissimilarity GCLM texture parameter from the HH polarisation backscatter |
| | HH_savg | Sum average GCLM texture parameter from the HH polarisation backscatter |
| | HV_contrast | Contrast GCLM texture parameter from the HV polarisation backscatter |
| | HV_diss | Dissimilarity GCLM texture parameter from the HV polarisation backscatter |
| | HV_intertia | Inertia GCLM texture parameter from the HV polarisation backscatter |
| | HV_savg | Sum average GCLM texture parameter from the HV polarisation backscatter |
| Landsat 8 | B2_median | Band 2 (blue) surface reflection median composite |
| | B3_median | Band 3 (green) surface reflection median composite |
| | B4_median | Band 4 (red) surface reflection median composite |
| | B7_median | Band 7 (shortwave infrared 2) surface reflection median composite |
| | B10_median | Band 10 (brightness temperature) median composite |
| | B11_median | Band 11 (brightness temperature) median composite |
| | bci_median | Biophysical Composition Index median composite |
| | ndbi_median | Normalized Difference Built Index median composite |
| | ndbai_median | Normalized Difference BAreness Index median composite |
| | ndvi_p10 | 10th percentile of the Normalized Difference Vegetation Index composite |
| | ndvi_median | Normalized Difference Vegetation Index median composite |
| | ndvi_p90 | 90th percentile of the Normalized Difference Vegetation Index composite |
| | ndwi_median | Normalized Difference Water Index median composite |
| Sentinel-1 | VV | Single co-polarization, vertical transmit/vertical receivemedian composite |
| | VH | Single co-polarization, horizontal transmit/horizontal receive median composite |
| | S1_StdDev | Standard deviation of VV and VH combined |
| Sentinel-2 | S2_ndvi_re1_median | Normalized Difference Vegetation Index Red Edge 1 median composite |
| | S2_ndvi_re2_median | Normalized Difference Vegetation Index Red Edge 2 median composite |
| Other | DEM | High accuracy global DEM at 3 arc second resolution from MERIT (Multi-Error-Removed Improved-Terrain Digital Elevation Model), version 1.0.3 |
| | DSM | Global digital surface model (DSM) dataset with a horizontal resolution of approximately 30 meters from ALOS World 3D, version 3.2 |
| | SLOPE | Slope derived from the MERIT DEM |
| | GCFH | Global forest canopy height data |
| VIIRS | NANTLI | Normalized Difference Vegetation Index adjusted Night-time Light Index, based on Landsat's NDVI and VIIRS Day/Night Band |

**Appendix G: Traditional accuracy assessment**

**Figure G1.** Overall and class-wise F1 accuracies for the global random forest LCZ models. Coloured boxes and grey whiskers span the 25-75 and 5-95 percentiles respectively. The means and medians are indicated by the white dots and black lines respectively.





# Appendix H: Thematic benchmark results per urban ecoregion


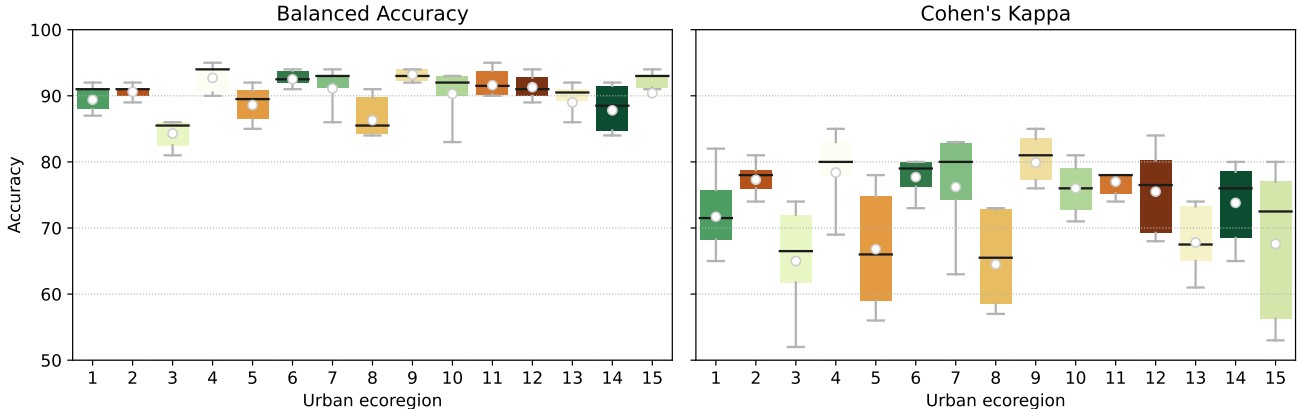

**Figure H1.** Balanced accuracy (left panel) and Cohen's Kappa (right panel) for the 150 global LCZ FUA regions in terms of built-up land, stratified by urban ecoregion (colours as in Fig. 3). Boxes and grey whiskers span the 25-75 and 5-95 percentiles respectively. The means and medians are indicated by the white dots and black lines respectively.



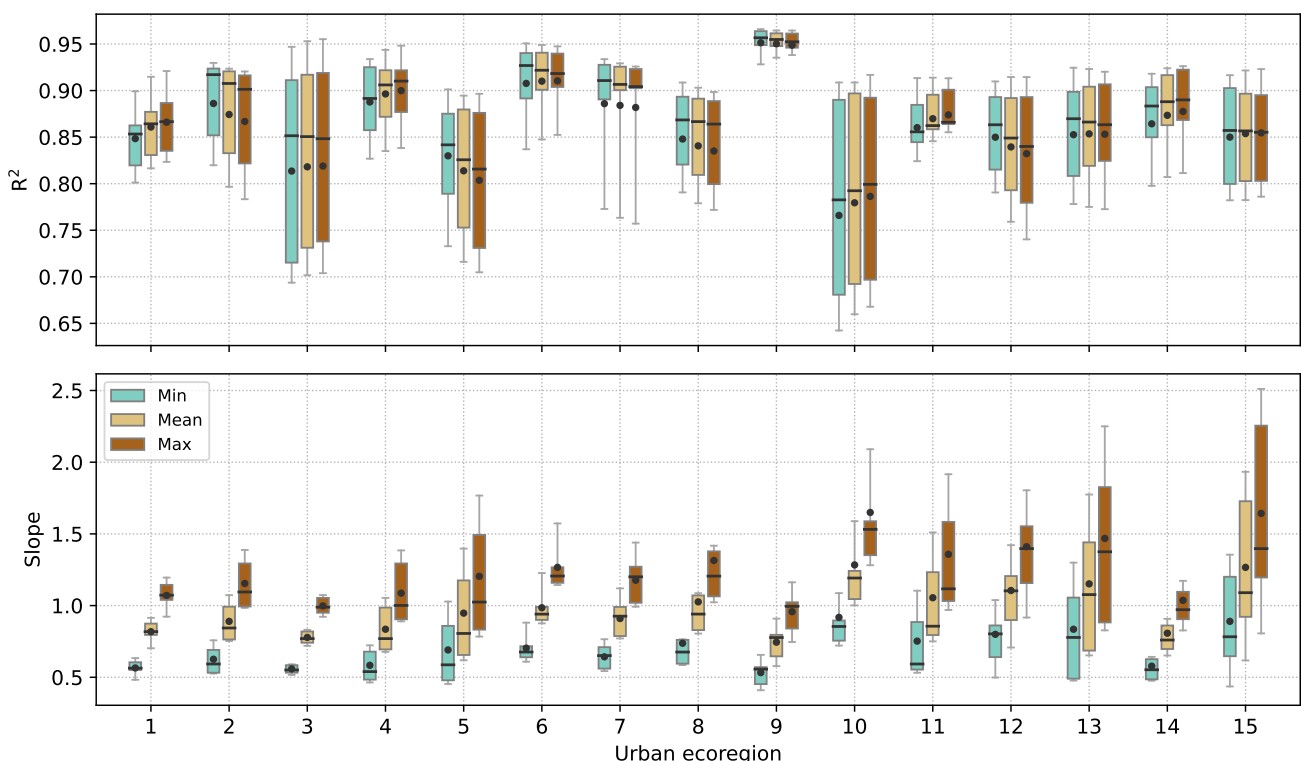

**Figure H2.** Coefficients of determination ($R^2$) and slopes resulting from the regression between the reference dataset for $\lambda_B$ and the corresponding universal LCZ-based values from Stewart and Oke (2012), using minimum, mean and maximum values (colours), for all FUAs stratified per ecoregion. Coloured boxes and grey whiskers span the 25–75 and 5–95 percentiles respectively. The means and medians are indicated by the black dots and lines respectively.

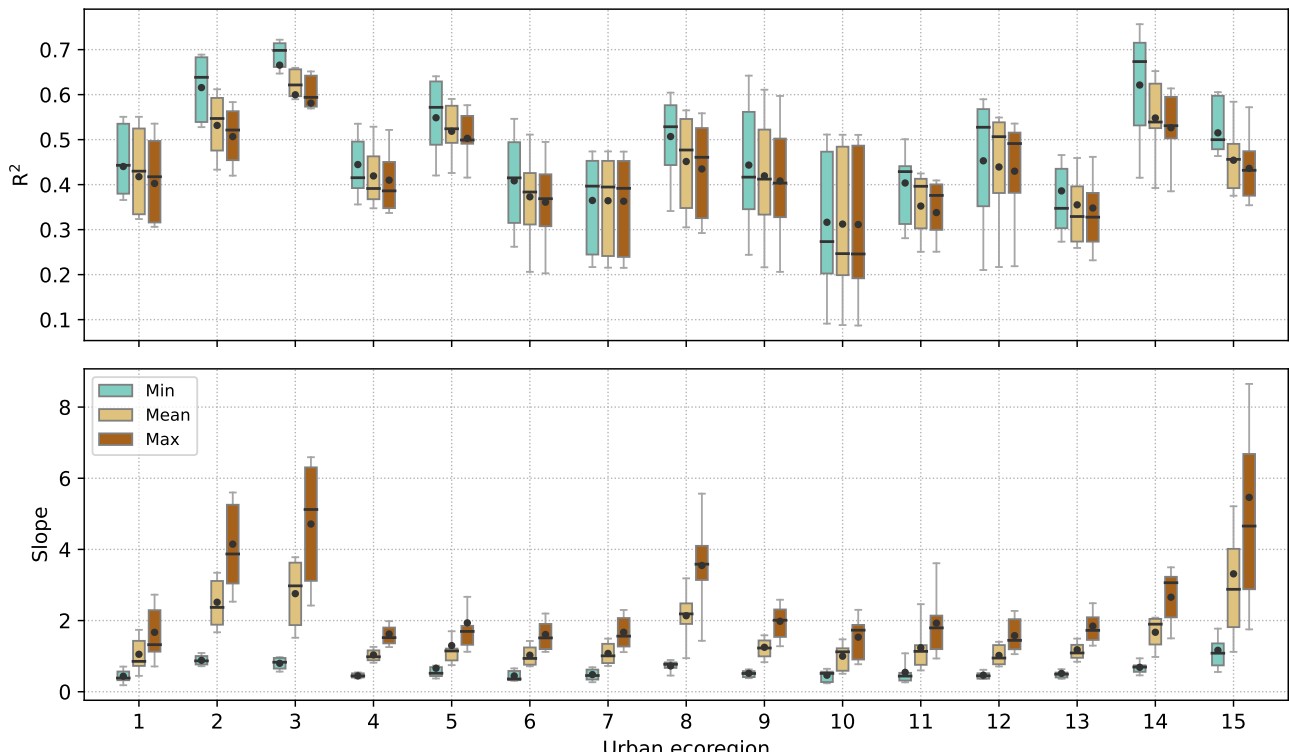

**Figure H3.** As Fig. H2 but for building height $H$.





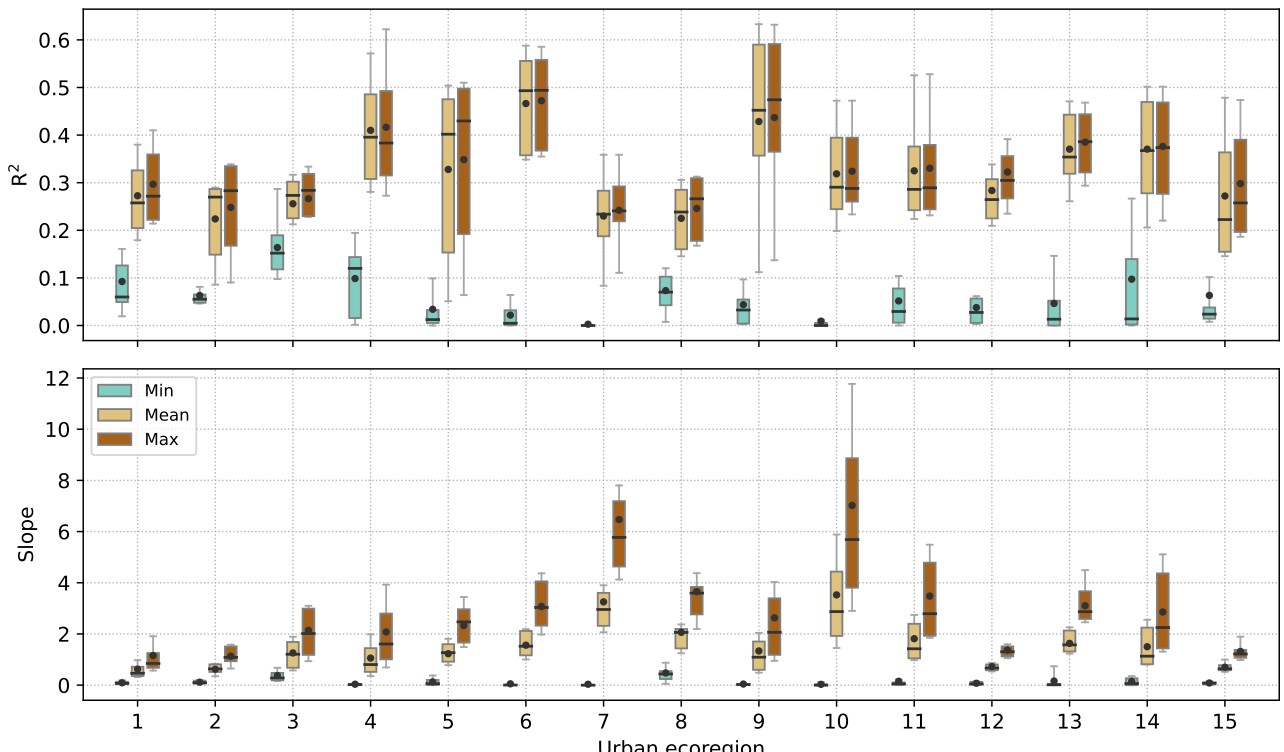

**Figure H4.** As Fig. H2 but for the anthropogenic heat flux $AHF$.