# Peer review of "A global map of Local Climate Zones to support earth system modelling and urban scale environmental science"

_Earth System Science Data, 2022_

## Author Comment (AC1)

**Manuscript essd-2022-92, First Revision**
**Submitted to: Earth System Science Data**

A global map of Local Climate Zones to support earth system modelling and urban scale environmental science

Demuzere, M., Kittner, J., Martilli, A., Mills, G., Moede, C.,
Stewart, I. D., van Vliet, J., and Bechtel, B.

**Response to Referee #1**

*July 14 2022*

We thank the chief editor and the reviewers for their appreciation of our work and the valuable comments on the manuscript. Please find the point-by-point responses below, indicated in blue.

**General comments**

The paper entitled "A global map of Local Climate Zones to support earth system modelling and urban scale environmental science" depicts the major advancements realised by the community effort led by the WUDAPT community that permitted the obtention of the first global Local Climate Zones map. The paper is already in a very mature state and no major points of concern are to be clarified. The paper should therefore be published once the comments given below are treated. Some specifications are required in the methods. But most importantly, I would like to have the authors commenting more on the quality of the training polygons and the sampling bias per ecoregion. This could help focus future efforts by the community to improve the current map in future releases.

Thanks for reviewing our research paper, and supporting its publication after addressing the comments described below.

**Major comments**

Line 120: Please define what a well-trained student is.

From previous works (e.g. Bechtel et al., 2015, 2017; Verdonck et al., 2019) we know that training of the human operator can improve the classification. This training includes an in-depth understanding of the LCZ scheme and its relevance in terms of urban climatology and thermal characterization of the urban environment, and an understanding of the guidelines that describe the optimal shape, size, frequency, siting, … of the training area (TA) polygons.

All of this context was discussed thoroughly with the RUB students. Afterwards they practised with the LCZ driving test and by creating TA sets for a large number of known cities that were reviewed by local experts. At the same time, the LCZ Generator capabilities were also used, allowing the students to revise the training areas based on the internal quality control, and subsequently re-submitting the revised training area files until an OA of at least 70% was reached. In this respect note that contributions from RUB members are not visible on the LCZ Generator platform, as we also use the system for internal procedures and tests.

So in short, the way of training our students is generally inspired by the outcomes of HUMINEX, which is indicated accordingly in the revised manuscript.

*Bechtel B, Alexander P, Böhner J, et al. Mapping Local Climate Zones for a Worldwide Database of the Form and Function of Cities. ISPRS Int J Geo-Information. 2015;4(1):199-219. doi:10.3390/ijgi4010199*

*Bechtel B, Demuzere M, Sismanidis P, et al. Quality of Crowdsourced Data on Urban Morphology—The Human Influence Experiment (HUMINEX). Urban Sci. 2017;1(2):15. doi:10.3390/urbansci1020015*

*Verdonck M, Demuzere M, Bechtel B, et al. The Human Influence Experiment (Part 2): Guidelines for Improved Mapping of Local Climate Zones Using a Supervised Classification. Urban Sci. 2019;3(1):27. doi:10.3390/urbansci3010027*

Line 127 to 129: How is the best submission defined? If solely on overall accuracy (OA), then this can be biased. Why not using all submissions instead, as suggested by the HUMINEX project. Explain why 50% is retained and not 60% as suggested by Bechtel et al. (2019). Please explain the rationale in a short sentence. Also, why are archived TAs given a higher priority over the ones produced in the Generator? Were they checked or published before being archived? Some places are mostly sampled via the Generator (e.g., India or China), would you say that the resulting mapping in these ecoregions are of lower quality?

For a first filtering, we indeed keep all submissions with OA => 50%. This is a balance between the guidelines of Bechtel et al. (2019) ("*a minimum average accuracy of 50% is required for each accuracy measure to pass the automated quality control before*"), and retaining enough samples on the global scale, in line with the findings of HUMINEX that "*poor to moderate quality TA sets can still contribute to good quality LCZ maps*".

As stated in the manuscript, archived TAs are "*collected from previously published research and collaborations, including the samples hosted on the old WUDAPT portal*". That means that most of these samples have undergone some sort of external quality control, and are hence given priority over those submitted directly to the LCZ Generator, for which we do not know whether or not they have been (externally) quality controlled.

On the China and India TA sets: I would not assume the resulting LCZ map for these regions is of lower quality. Many of the Indian sets were actually double-checked by some of the co-authors, and TA polygons were often labelled as suspicious, mostly because of their shape or size not being according to the guidelines. These polygons are removed via the shape/size filter as mentioned in the manuscript: "*too small or too complex TA polygons are removed*".  Many of the Chinese TA sets submitted to the LCZ Generator are actually also used in peer-reviewed publications, and as such have undergone some sort of external review as well.

The above does of course not remove all uncertainty embedded within the TA sets from the LCZ Generator, but also here we assume that their volume (these areas have relatively much more TA sets compared to other ecoregions) serves the wisdom of the crowd idea (Bechtel et al., 2017, Verdonck et al., 2019), still resulting in good quality LCZ maps.

*Bechtel B, Alexander PJ, Beck C, et al. Generating WUDAPT Level 0 data – Current status of production and evaluation. Urban Clim. 2019;27:24-45. doi:10.1016/j.uclim.2018.10.001*

Line 169 to 174: You say it in the following paragraph but it is unclear at this stage why you do the feature importance for the 16 sets of TAs. Also, why is the performance measured at that stage and not in GEE? Could you also be a bit more specific on the reasons that explain you going from GEE to a python environment? Could be interesting for some geospatial scientists.

Since the feature importance results of the 16 spatial regions are only used in Pathway 2, we believe it is appropriate to keep that information in that section only, and not already to describe its purpose in the Pathway 1 section.

The main reason for doing Pathway 1 offline is provided in the original Line 164, referring to the sheer size of the classification problem (2+ million labels and 46 input features). Trying to solve this in EE results in exceeding the available user memory limit or leads to a computation time out. This information is now added to the revised manuscript.

Line 184 to 185: I like that step and fully support it. Nonetheless, there may ba a bias induced by the quality of the TA per ecoregion (e.g., TAs comming only from the Generator). This should be discussed at a certain point.

To clarify: for each of the 5 seed iterations, 10% of all selected TA sets are used, that are balanced across LCZ class labels and ecoregions. That means that in reality, each subset will be composed out of a mixture of global RUB, ARC and GEN TA samples. These subsets are subsequently used to make one global LCZ map, which is then repeated 50 times (5 seeds, 10 iterations per seed). Given this mixture of types of TA sets in each iteration, we don't think this will introduce a bias in the LCZ map quality. We clarified in the manuscript that TA sampling is done from all selected TA sets.

Line 192 and 193: I had a hard time understanding why you calculate the accuracy again for each subset after going through Pathway 1.

The accuracies from Pathway 1 are obtained using all 46 original input features and a type of TA sampling. As the final LCZ map is produced in Pathway 2, using only 30 input features and a different type of TA sampling, we want to make sure that the final accuracy assessment matches the underlying procedure that was used to create the global LCZ map. Some notes have been added in the text to make this difference more clear.

Line 249: Do you know how the GHS-S2net data performs in places where informal settlements are common and where roads are made of bare soil rather than asphalt? This could impact your evaluation.

As far as we know, there are no global high-resolution layers on building surface fraction and imperviousness. Because of this, we only assessed the representativity of the GHS-S2net data for these properties over Europe, using EEA's datasets. That of course excludes certain types of urban forms that do not exist in Europe.

Yet this benchmark with EEA data was merely an additional test, to double-check the findings of Corbane et al. (2021), who state that *there is a strong relationship between the GHS-S2net output probabilities and the building densities*. Their findings are based on using building footprints from 277 regions across the globe. The latter reference database is described in more detail in Corbane et al. (2019): *"It is a reference spatial database including single building delineation (more than 40 million individual building polygons) derived from digital cartography, at a nominal scale of 1:10,000, compiled to have the most possible representative sample set from different cities around the globe"*.

From Figure R1 below one can see that, even though the majority of building footprints are sampled in developed regions, the dataset also contains information from less-developed countries. But since the GHS-S2net data is found to be a proxy for building footprints, and is used as such in the manuscript, we don't think this has an impact on our evaluation.

Note that the Corbane et al. (2019) reference is added in the revised manuscript to more clearly point to the characteristics of the reference data.

[Figure]

**Figure R1**. Location of the 277 Areas of Interest used in the validation and the number of building footprints within each of them (Corbane et al., 2019, their supplementary File 2, Figure 1).

*Corbane C, Pesaresi M, Kemper T, et al. Automated global delineation of human settlements from 40 years of Landsat satellite data archives. Big Earth Data. 2019;3(2):140-169. doi:10.1080/20964471.2019.1625528*

*Corbane C, Syrris V, Sabo F, et al. Convolutional neural networks for global human settlements mapping from Sentinel-2 satellite imagery. Neural Comput Appl. 2021;33(12):6697-6720. doi:10.1007/s00521-020-05449-7*

Line 288 to 290: Looking at the TAs on the LCZ Generator, one can see that in the Indian cities, for example, close to no LCZ 7 has been sampled. Coming back to the question of the TA quality in certain places, how do you think this could influence your global map? Also, could it be that some users do not take sufficient time to get acquainted with the LCZ scheme? Would your TA filtering capture this?

As can be seen from Figure E1, LCZ class 7 indeed has the least amount of polygons, across all urban ecoregions. It is therefore likely that this LCZ class is under-represented in our TA database, which of course has repercussions for the representation of this LCZ class in the global LCZ map. This is also identified within the manuscript, by eg. pointing to the lowest LCZ probabilities for this class (Lines 359-360 in original manuscript). More in general, the representation of LCZ 7 in the global map is highlighted as the first limitation ("*For example, LCZ 7 requires more attention and alternative mapping strategies*" - Line 509 in original manuscript), and we agree that this class needs further attention in the future, as stated in Lines  363-365: "*For this LCZ type, future versions of the global map might benefit and built further upon recent efforts dedicated to map informal urban settlements (see e.g. Kuffer et al., 2020; Assarkhaniki et al., 2021; Owusu et al., 2021; Abascal et al., 2022)."*

As mentioned previously, it is clear that our TA filtering procedure can not remove all uncertainty embedded within the TA sets. But we do believe that the volume of training pixels used in the training of the LCZ classification models allows for the development of a good quality global map.

Figures 5, 6 and 7 and related text: I would like the authors to comment more about the probability of a certain LCZ to occur in different FUAs. In Lagos, for example, the probability of having the same LCZ classified is higher in the city and lower in the rural area. This is the opposite for a city like Delhi or Lima. Could you try to explain and discuss how the quality of the TA sampling done in the different ecoregions may lead to such outcome?

For clarity, the probability does not indicate the probability of a certain LCZ to occur in a certain FUA. Yet is it the number of times the 50 RF models mapped the modal LCZ, serving as an indicator of the robustness of the LCZ class label for a particular pixel. The meaning of this probability layer is included in the text: "*In addition, a classification probability layer is produced that identifies how often the modal LCZ was modelled per pixel (e.g. a classification probability of 60\% means that the modal LCZ class was mapped 30 times out of 50 LCZ models)*". However, in order to avoid confusion, we have changed all occurrences of "*probability*" to "*classification probability*", to clarify that this is not the probability of a LCZ label to occur somewhere, but the probability of the 50 RF models to classify the final modal LCZ.

The classification probability maps that are shown in Figs. 5-7 are discussed in more detail in Section 3.2 and Figure 9. We do think that the latter section and figure already provides a good overview of the robustness of each mapped LCZ class, per urban ecoregion. For example, for LCZ 2, the mean classification probability over all 13000 FUA's is ~60%, with mean values to vary between ~50% (ER6) and ~70% (ER12) when stratified according to urban ecoregions. Or for LCZ D: ~75% (global), ~65% (ER7) and ~82% (ER8).

We believe that on this scale, it is not very informative to discuss differences between individual LCZ classes, FUAs and/or ERs. What this section 3.2 and Figure 9 however describes is that "*all classification probabilities per LCZ class are in line with the global values, demonstrating the universality of the LCZ typology and the robustness of the classifiers and input features across the urban ecoregions*". It also indicates areas where more work is needed, such as the mapping of LCZ class 7, as discussed above. Also, given the current balanced sampling across all selected TAs - with a mixed set of TAs informing every single classifier - we don't think this strategy influences the quality of the map. Instead, these classification probabilities are more indicative of how difficult it is for the classifiers to recognise specific LCZ classes (e.g. LCZ 6 performs better than LCZ 7, or LCZ A better than LCZ C), providing directions for future work.

Finally, note that, as requested by the reviewer, Fig. 9 is also displayed as a function of ER instead of LCZ class. See below, or Figure H1 in the new Appendix H of the revised manuscript.

Line 331: Does the LCZ 8 class really belong in this cluster? Shouldn't it be added to the group with LCZ 7 and 10? Afterall, the building materials of LCZ 8 are very different to the compact built-up LCZs.

As stated in the manuscript (Lines 330-331), the grouping is first of all done *according to their degree of total impervious fraction*. As such we decided to put LCZ 8 in the HIGH-$\lambda_T$ cluster.

Line 339: Although I do believe that LCZ 3 and LCZ 8 are indeed the most common LCZ globally, the proportion of LCZ 8 over LCZ 3 may be biased because some confusion is happening during the classification. Could you try to explain why such confusion is happening between these two classes? You later speak about their radiative ressemblance (on line 511). Do you have any data to support this?

Since the LCZ map is the product of a machine learning technique, all LCZ classes are subject to uncertainty, which is indicated by the sections dealing with accuracy assessment in general, and the information on the LCZ classification probability in particular. So all LCZ labels will be subject to confusion, as is the case for any (global) product that is the result of an automated algorithm (or even manual classification).

According to Figures 9, G1, and H1 - looking at the built LCZs only - the accuracies and classification are highest for LCZ 3 and 8 (and LCZ 6), indicating that the classifiers provide robust results for these classes. But even here, the classification is not perfect, and confusion might still

occur. This is especially visible for the LCZ map of Lima in Fig 7, that shows too large an extent of the city being mapped as LCZ 8, which in reality should be mapped as LCZ 3.

In order to understand the spectral characteristics of each LCZ class, we've done an extensive qualitative assessment of their spectral profiles (analogues to the analysis done in Demuzere et al., (2019), their Figure 2) during the model development (not shown in the manuscript). This helped to understand the results of the automated EO input feature importance and the RF models. See eg. Figure R2, that depicts the normalised input feature profiles for the top 10 input features for LCZs 3, 8 and the range for all built LCZs (1-10). Even though this is just a snapshot of the big data underlying the RF models, it does show that often LCZs 3 and 8 have similar spectral profiles, with value ranges that are often only a fraction of what is available amongst all built LCZ classes. As we think this information is not critical to the paper, and given that the manuscript already contains a large number of Figures and Appendices, we prefer not to elaborate on this in the manuscript. We did however rephrase this sentence in the manuscript to make it more clear: "*Confusion may also exist between classes with similar impervious and built-up surface fractions, characterised by similar spectral characteristics (not shown), which can lead to confusion between these classes (see e.g. the confusion between LCZs 3 and 8 in the LCZ map for Lima (Peru, ER11), Fig 7)*"

[Figure]

**Figure R2**: Normalised spectral profiles for LCZ classes 3, 8 and all built classes (1-10), extracted for all 2M+ TA pixels, for the top 10 earth observation input features (see Figure F1). Boxes and whiskers span the 25–75 and 0-100 percentiles respectively. The medians are indicated by the orange lines.

*Demuzere M, Bechtel B, Mills G. Global transferability of local climate zone models. Urban Clim. 2019;27(November 2018):46-63. doi:10.1016/j.uclim.2018.11.001*

Figure 8: I really like this figure but could you add an estimation of the uncertainty of the proportion per LCZ?

For the sake of clarity and readability, we'd prefer not to add an additional layer of information to this figure. In case a user is interested in the robustness of each LCZ class label per ER, he/she can interpret Figure 8 alongside Fig. 9.

Line 521: When you talk about "their purpose", could you add that users are invited to continue helping the develoment of future maps releases by contributing to the WUDAPT project through the LCZ Generator?

Thanks for this suggestion. We have added the following statement to this section: "*In addition, interested users are invited to actively contribute to future releases of this product, by submitting city-specific training area sets to the LCZ Generator. This community engagement will not only improve the quality of next LCZ map releases, but also contributes to the overall WUDAPT philosophy to provide urban canopy information and modelling infrastructure to facilitate urban-focused climate, weather, air quality, and energy-use modelling application studies (Ching et al., 2018)*".

**Minor comments**

Line 2: Change "as" to "since" and "acknowledged" to "recognized"

We have rephrase this sentence to the following:

*This data can support a range of environmental services, since cities are places of intense resource consumption and waste generation, and of concentrated infrastructure and human settlement exposed to multiple hazards of natural and anthropogenic origin.*

Line 6: Add "and mitigative role" at the end of the sentence

Added.

Line 19: Change "warming" to "climate warming"

Changed.

Line 19 to 20: Rephrase this complex sentence and potentially divide it in two to make it clearer

Unchanged, as we believe this sentence is clear.

Line 34: Change to "and alters the local climate creating specific urban climates" or similar.

We prefer to keep the original sentence, keeping the general focus of altering the "urban climate".

Line 44: Chose between "distinct urban canopies and boundary layers" or "a distinct urban canopy and related boundary layer".

Changed to *… creating distinct urban canopy and boundary layers.*

Line 46 to 47: I would remove this statement that is not defended by any evidence. Otherwise, put it subjectively (e.g., "could" soon allow; "are expected"...)

Variable-scale (-or resolution) model systems already exist for a while, see eg. Haung et al. (2016) as one of the pioneering studies with a global model, or the more recent work done at CSIRO (here). Providing more details in the intro is outside the scope of the study, but we do believe it is fair to keep the current statement.

Huang X, Rhoades AM, Ullrich PA, Zarzycki CM. An evaluation of the variable-resolution CESM for modeling California's climate. J Adv Model Earth Syst. 2016;8(1):345-369. doi:10.1002/2015MS000559

Line 48 to 49: Rephrase as "Hence, a comprehensive [...] is needed."

We believe that "What is needed …" fits better with the structure of the full sentence

Line 53: Change "needed to support" to "required by" and change the final dot to a double point "[...] functions: measures of [...]".

The original sentence is kept, as otherwise the logic of first describing "Measures of form…", and afterwards "Urban functions …", is lost.

Line 55: "Influences" to "Influence"

Changed.

Line 60: Change "assess" to "test"

We believe "assessment" is the proper term to describe the benefits of climate-based interventions.

Line 64: Add a space between "heat" and "(Demuzere"

Changed.

Line 77: Add "[...] parameters (UCPs) required by urban climate models and by policy-makers to run [...]"

We prefer to keep the original sentence, as we believe that - in general - urban policy-makers do not run models to make informed decisions.

Line 96: Check the citation command for Ching et al. (2018). If LaTeX used, check that for all the manuscript.

Thanks for identifying this typo. This is adjusted.
Since we indeed use LaTeX, the manuscript is checked for the proper use of \cite{} and \citep{} throughout the manuscript.

Line 108: Change "random forest model" to "random forest classifier".

Changed.

Line 138: Rephrase "one needs" to a less familiar tone

Changed.

Line 164: "2+ million labels", are these TAs or pixels within TA polygons?

Pixels; so individual labels for the RF classifier.
For the final classification, 63847 polygons are used, as indicated in the beginning of the results section.

Line 171: Delete the comma after "a)"

Changed.

Line 195 to 196: Is the "splitting the polygon pool" approach done for the first time in the LCZ mapping or has it been used in previous mapping (e.g., Europe or the US)?

So far, various techniques have been applied. The vast majority of accuracy assessment splits individual pixels. Some others, eg. Demuzere et al. (2020), also performs an accuracy assessment by splitting the TA sample in terms of cities (all-but-one city approach).  To the best of our

knowledge, the work of Xu et al. (2021) is the first to explicitly use this "splitting the polygon pool" approach.

Line 291 to 292: Please detail what the "average number of ROIs" is.

"Number" refers to "amount". This has been changed.

Line 322 to 325: This sentence could be moved to the discussion if needed. Otherwise, please suppress it.

This sentence is condensed, just keeping the example of the UHI characterisation.

Figure 9: Could you provide boxplots per ER too?

Thank you for this suggestion. We have created this Figure R3, and added it to the Appendix (Fig. H1), and referenced it in the section where the probabilities are discussed.

[Figure]

**Figure R3**. Classification probabilities of the mapped LCZ classes, aggregated over all urban centres from GHS-UCDB. The grey boxplots depict the probability distribution for all global urban centres, per ER, with boxes and whiskers spanning the 25-75 and 5-95 percentiles respectively, and means and medians indicated by the white dots and black lines respectively. The vertical lines in the colours of the LCZ classes indicate the 25 to 75th percentile range averaged over the urban centres, with LCZ-colored dots indicating the mean.

Line 388: Why is the slope chosen as a metric for evaluating the classification performance? This is quite uncommon.

We chose to use the slope metric, inspired by the assessment done in Corbane et al. (2021), where they used this metric to assess the quality of the output probabilities in terms of the building footprint reference dataset. It is an easy-to-understand metric that provides a rapid assessment of the predictive power of LCZ-based urban canopy parameters compared to their observed counterparts.

*Corbane C, Syrris V, Sabo F, et al. Convolutional neural networks for global human settlements mapping from Sentinel-2 satellite imagery. Neural Comput Appl. 2021;33(12):6697-6720. doi:10.1007/s00521-020-05449-7*

Line 401 to 403: How is this statement explanatory of the difference between the LCZ-derived AHF and the observation?

We do think that Lines 400-414 provide sufficient context of why AHF is used in the first place (lack of global datasets on thermal and radiative properties of the urban fabric), and why one can not

expect that the generic mean annual AHF value provided by Stewart and Oke (2012) is able to capture the "observed" global variability in AHF, especially in terms of its zonal variation.

Line 407: Chose another word than "zonal"

Zonal is commonly used to refer to variations across latitudes, as is also done in Varquez et al. (2021), their Fig. 5. As such the term is kept here.

Line 422: Please change "Global South" and later "Global North" to other denominations. This concept dates from the 1980s.

This is a valid concern, but we basically just adopted the terminology of the paper that performed this research (Nagendra et al., 2018). We have changed this now to low, middle, and high income countries, a terminology used by the Worldbank.

*Nagendra H, Bai X, Brondizio ES, Lwasa S. The urban south and the predicament of global sustainability. Nat Sustain. 2018;1(7):341-349. doi:10.1038/s41893-018-0101-5*

Line 441 to 442: Do you have a reference to defend that city population is a proxy to urban form?

To clarify, this is not a practice we "defend". It is merely an observation of what is done elsewhere. We have added some references that use this approach.

Line 453: The works by Potgieter et al. (2021) and Brousse et al. (2022) are suggested as additional references concerning crowdsourced data.

Added.

Line 471: When citing Demuzere et al. (2021a), please refer specifically to the W2W python tool as done for WUDAPT-TO-COSMO.

Adjusted.

Line 510: I suggest changing "surface fractions" to "impervious and built-up surface fractions".

Adjusted.

Line 512: Rephrase this sentence for clarity.

This sentence changed to: "*Confusion may also exist between classes with similar impervious and built-up surface fractions, characterised by similar spectral characteristics (not shown), which can lead to confusion between these classes (see e.g. the confusion between LCZs 3 and 8 in the LCZ map for Lima (Peru, ER11), Fig 7)*"

Please consider checking for American and English spelling discrepancies.

We have removed all American / UK English discrepancies.

---

## Author Comment (AC2)

**Manuscript essd-2022-92, First Revision**
**Submitted to: Earth System Science Data**

A global map of Local Climate Zones to support earth system modelling and urban scale environmental science

Demuzere, M., Kittner, J., Martilli, A., Mills, G., Moede, C.,
Stewart, I. D., van Vliet, J., and Bechtel, B.

**Response to Referee #2**

*July 14 2022*

We thank the chief editor and the reviewers for their appreciation of our work and the valuable comments on the manuscript. Please find the point-by-point responses below, indicated in blue.

**General comments**

This work describes a new dataset of land cover types (10 urban – 7 natural) using Local Climate Zones at the global scale. Work is clearly described, evaluated, and presented. The associated dataset is of high quality, and I expect will become a landmark data source for the community.

Thanks for reviewing our research paper, and supporting its publication after addressing the clarifications described below.

The discussion on "accuracy" vs "robustness" could be improved (see specific comments). Additionally, there is no acknowledgment that LCZ training polygons are susceptible to human errors (again see specific comments).

See responses below for details on "accuracy versus robustness".

On the last point. In Section 2.1 we state that "*While the LCZ maps created by individuals are often of poor to moderate quality, The Human Influence Experiment (HUMINEX) (Bechtel et al., 2017; Verdonck et al., 2019a) demonstrated large accuracy improvements (up to 20%) when multiple (poor to moderate quality) training datasets were used together to create a single LCZ map*".

To clarify, we have extended the first part of this sentence to: "*While the training area polygons and corresponding LCZ maps created by individuals are often of poor to moderate quality …*". We believe this statement is appropriate, not only referring to potential errors introduced by the subjective interpretation of the human operator, but also the consequences of perception, interpretation, experience and prior knowledge (see HUMINEX for more details).

Section 3.2 and Figure 10 show that the correlation R2 for building height is only ~0.5, however this is only very briefly mentioned in results, and not mentioned elsewhere (e.g. discussion/conclusion/abstract). So, while 2D information like lambda_B appears to be very well captured, 3D information remains a significant limitation. This is a key result and its implications should be discussed more thoroughly.

Thanks for this suggestion. We have added some notes on the 2D versus 3D information in the discussion (last paragraph), as follows:

" *… More in general, the results of the thematic benchmark reveal that two-dimensional informa-*

*tion (urban land cover and building surface fractions) is well represented, but that the corresponding three-dimensional (3D) information requires more attention. Ongoing developments such as the work on the Digital Synthetic City Ching et al. (2019), tailored towards providing more detailed information on the urban landscape (WUDAPT Levels 1 and 2), or global 3D building information (Li et al., 2020a; Esch et al., 2022; Kamath et al., 2022) might contribute to improve the quality of future LCZ map release…. "*

A lower reliance on acronyms would assist the casual reader. For example Figures 9 and 10 are not decipherable without referring to other sections of the text.

Thank you for this suggestion. We have revised the captions of all figures and tables in the main manuscript, and adjusted the text where needed, to make sure its meaning is clear without the need to refer to other sections of the text.

However, overall, an impressive body of work.

**Specific comments**

Line 41: "Earth System Models (ESMs) have only recently evolved to accommodate urban-scale landscapes, even though the parameters that are used by ESMs to these landscapes are limited in scope"

Some global climate models have had integrated urban canyon models for over a decade (e.g. CLMU in CESM). I'm not sure if these are ESMs (ESM relates to the carbon cycle, not the global scale, some readers may misinterpret this). I think safer/clearer to say many global-scale models ignore urban landscapes or represent them simply.

The context of this statement is that in CMIP5, only one of the GCM/ESM models was dealing with urban surfaces (CESM with the Community Land Model (CLM) as land surface model (LSM)). In the most recent CMIP6 archive, there are a few more, all of which use CLM as LSM. So in that sense we believe that the statement on the fact that models only recently (given the long history of global-scale climate models and their LSMs, Fisher and Koven (2020)) evolved to accommodate urban landscapes is accurate. Yet we agree the distinction between GCMs and ESMs might be misinterpreted by some of the readers, so we have adjusted this sentence accordingly.

*Fisher RA, Koven CD. Perspectives on the Future of Land Surface Models and the Challenges of Representing Complex Terrestrial Systems. J Adv Model Earth Syst. 2020;12(4). doi:10.1029/2018MS001453*

Line 120: suggest removing "well-trained" as subjective.

From previous works (e.g. Bechtel et al., 2015, 2017; Verdonck et al., 2019) we know that training of the human operator can improve the classification. This training includes an in-depth understanding of the LCZ scheme, its context in terms of urban climate and thermal characterization of the urban environment, and an understanding of the guidelines that describe the optimal shape, size, frequency, siting, … of the training area (TA) polygons.

All of this context was discussed thoroughly with the RUB students. Afterwards they practiced with the LCZ driving test and by creating TA sets for a large number of known cities that were reviewed by local experts. At the same time, the LCZ Generator capabilities were also used, allowing the students to revise the training areas based on the internal quality control, and subsequently re-submitting the revised training area files until an OA of at least 70% was reached. In this respect note that contributions from RUB members are not visible on the LCZ Generator platform, as we also use the system for internal procedures and tests.

To summarize, we would like to keep the "well-trained" formulation in the text, as we believe this is an important part of the LCZ mapping workflow. Since the way of training our students is generally inspired by the outcomes of HUMINEX, this is indicated accordingly in the revised manuscript.

*Bechtel B, Alexander P, Böhner J, et al. Mapping Local Climate Zones for a Worldwide Database of the Form and Function of Cities. ISPRS Int J Geo-Information. 2015;4(1):199-219. doi:10.3390/ijgi4010199*

*Bechtel B, Demuzere M, Sismanidis P, et al. Quality of Crowdsourced Data on Urban Morphology—The Human Influence Experiment (HUMINEX). Urban Sci. 2017;1(2):15. doi:10.3390/urbansci1020015*

*Verdonck M, Demuzere M, Bechtel B, et al. The Human Influence Experiment (Part 2): Guidelines for Improved Mapping of Local Climate Zones Using a Supervised Classification. Urban Sci. 2019;3(1):27. doi:10.3390/urbansci3010027*

Line 127: "only the best submission is retained" what distinguishes a "best" submission?

With "best" we refer to the submission of the same city with the highest overall accuracy. This is clarified in the text accordingly.

Line 128: How is accuracy determined?

During the filtering process, we mostly use the overall accuracy (OA) metric provided by the LCZ Generator (Demuzere et al., 2021) as a guideline to select good submissions, which is partly in line with the recommendation of Bechtel et al. (2019). But as also written on the FAQ of the LCZ Generator (see here), we do acknowledge the fact that high overall accuracies do not automatically mean that the map is correct, or that all TA polygons are a correct representation of the landscape. Yet dealing with such big data (80000+ polygons before filtering) requires automation procedures, meaning that compromises have to be made to come up with a workable solution.

*Bechtel B, Alexander PJ, Beck C, et al. Generating WUDAPT Level 0 data – Current status of production and evaluation. Urban Clim. 2019;27:24-45. doi:10.1016/j.uclim.2018.10.001*

*Demuzere M, Kittner J, Bechtel B. LCZ Generator: A Web Application to Create Local Climate Zone Maps. Front Environ Sci. 2021;9. doi:10.3389/fenvs.2021.637455*

Section 2.4.1: I would describe this as a test of robustness, not accuracy, as this does not test whether the classifications are correct, just whether they change with different inputs. This method also assumes that training areas are accurate, but TAs are classified subjectively by humans. True accuracy can be tested with building resolving spatial datasets. However, I accept this "accuracy" terminology has been established elsewhere in the literature, but a comment to clarify accuracy vs robustness would assist readers.

It is a general problem in LCZ mapping that there is typically no independent testing data available and also there is not even necessarily one true class for each pixel as discussed in Bechtel et al. (2015). Thus we use two independent approaches to test the quality of the product - a comprehensive cross-validation scheme and a thematic benchmark. Thus the accuracy measures always reflect a comparison between the classification result and independent samples. We agree that the metrics are well established and we thus prefer to keep this terminology. We do however provide some more context on the meaning of the accuracy metrics, by adding the following statement in the revised manuscript:

*"It is important to note that these accuracy metrics reflect the consistency of the TA samples, but do not guarantee that the TA polygons are semantically correct. However, since a huge TA database from various sources and cities was used, this gives much more confidence than using a TA set for a single city."*

Line 200: "The overall accuracy denotes the percentage of correctly classified pixels." As described above, the method does not assess whether pixels are classified correctly, only how often they are unchanged (and potentially remain incorrect). With poor training data, the overall "accuracy" could approach 100% but be completely wrong. Please rephrase.

100 % accuracy could only be achieved using a single class as training data, which is excluded in the given procedure using balanced training samples. Yet it is true that the measure is based on independent sample data which also contains errors. This was added to the description:

"*The overall accuracy denotes the percentage of independent test pixels that were assigned the same class as the test label. $OA_u$ reflects this percentage for the urban LCZ classes only, and $OA_{bu}$ is the overall accuracy for the built versus natural LCZ classes only, ignoring their internal differentiation.*"

Line 422: While the use of "Global South" and "Global North" is quite common, some see these terms as problematic as they are geographically inaccurate, deterministic, and paternalistic. If authors mean "lower wealth" they could just say that.

This is a valid concern, but we basically just adopted the terminology of the paper that performed this research (Nagendra et al., 2018). We have changed this now to low, middle and high income countries, a terminology used by the Worldbank.

*Nagendra H, Bai X, Brondizio ES, Lwasa S. The urban south and the predicament of global sustainability. Nat Sustain. 2018;1(7):341-349. doi:10.1038/s41893-018-0101-5*

**Technical corrections**

None

---

## Author Comment (AC3)

**Manuscript essd-2022-92, First Revision**
**Submitted to: Earth System Science Data**

A global map of Local Climate Zones to support earth system modelling and urban scale environmental science

Demuzere, M., Kittner, J., Martilli, A., Mills, G., Moede, C.,
Stewart, I. D., van Vliet, J., and Bechtel, B.

**Response to Referee #3**

*July 14 2022*

We would like to thank Dr. Jason Ching for his extensive appreciation of our work and the valuable comments on the manuscript. Point-by-point responses to the minor issues are mentioned below, indicated in blue.

**Preface to this review**: A decade ago at the Croucher Advanced Study Institute in Hong Kong, this Reviewer and Gerald Mills (a coauthors of this paper) reflected upon a presentation by Iain Stewart (also co-author on Local Climate Zones (LCZ), topic of his PhD research. The LCZ is a universal classification scheme that differentiates urban surfaces into different combinations of building form and function features. Together with associated values of urban canopy parameters (UCPs) the LCZ provided the conceptual framework that inspired the startup of WUDAPT, an urban climate community collaborative project. WUDAPT scope is worldwide, in principle the global LCZ map with its companion UCPs provides model inputs that describe the underlying embedded canopy features of the urban boundary layer for any and all cities in the world. This paper describes efforts leading to the generation of the Global LCZ map, an achievement that culminates an effort a decade in the making and satisfies the major goal and is a key milestone of the WUDAPT. This global LCZ map product support "fit for purpose" applications of environmental models capable of addressing urban induced environmental issues exacerbated by climate changes.

**Overview**: This paper is a significant contribution; it represents the product at 100-meter resolution designed to provide urban canopy data, making possible a means to generate uniformly consistent urban canopy parameters for all cities in the world as inputs to a wide variety of models such as meteorology (e.g., WRF_ Surface energy budgets (e.g., SUEZ), etc. The implication of this result is a capability for making possible fit for purpose (FFP) modeling for assessment at intra-urban scales impacted by climate changes for any and all cities in the world. The LCZ scheme is the cornerstone of this paper; it defines 10 distinct classes each having unique land cover and physical form and function aspects describing the built environment along with 7 other nonurban land cover types. For each of the built classes, there is a corresponding range of values of urban canopy parameters suitable as modeling inputs for urbanized WRF and other environment modeling systems. The research team has been engaged in developing and establishing methods and techniques at the outset of WUDAPT; their creating of an LCZ generator facilitated upscaling LCZ maps based on sets of Training Areas (TA) representing each LCZ class by urban experts for individual cities upscaled to regional/continental scale maps for different regions of the world. Their R&D trajectory has provided the experiential base and creating approaches and methods that extend TA transferability from individual cities to regional and continental maps. This paper describes in detail the adjustments and modifications to the methodology that generated regional-continental LCZ maps thus enabling the creation of this global LCZ map. This achievement fulfills a major objective of the WUDAPT (www.Wudapt.org) Project.

**Key Points**: The article provides the reader with (i) a concise discourse describing the approach and methods and inputs to generate this product; and (ii) provide suggestions on its utility to supporting modeling and urban scale environmental science. The context of my review will reflect perspectives based on the WUDAPT initiative, a collaborative venture of the urban community. This article is thoughtful, and well organized, I briefly highlight and summarize key points of each section below.

**The Title** is accurate and appropriate. The focus of this paper in on the development and implementation details of the methodologies that makes possible the resulting global LCZ map. LCZ is an universal urban typology and classification scheme to representing unique properties of form and function variables in the urban canopy layer of cities along with 7 other classes representing non-urban landscapes.  The complementary urban canopy parameters values associated with each LCZ class provide a globally consistent framework for data on the form and function of morphological features in urban canopies. This effort extends LCZ maps progressing from original city specific sets to recent mapping of cities and the surrounding non-urban areas within regional domains to global coverage. The impact of this achievement supports urban scale modeling systems and their applications anywhere in the globe, achieving the primary objective of the Level 0 approach of WUDAPT. Implementation procedures to generating the upscaled Global product required appropriate modification and innovations to previous efforts for the regional scale prototypes.

**The Introduction** provides a perspective on the value and importance of urban science to addressing global climate change issues. The enabling science behind current models is the physics algorithms for the vertical exchange processes of momentum, energy and moisture pollutant emission influenced by the myriad of urban morphological (UM) features. These exchange processes take place in the so-called urban canopy layer which extends from the surface to the top of UM features. Typically, modeling with urban canopy physics requires special sets urban data but heretofore is only available for limited number of cities; thus a large information gap exists which severely limits environmental models as tools intended and needed to addressing climate induced risks issues at urban scales. The introduction of the LCZ framework and its companion range of values of urban canopy parameters (UCPs) is what the Global LCZ map achieved here and generated at 100m resolution makes possible myriad of practical "fit for purpose" environmental modeling applications at a reasonably fine scale addressing climate change issues impacting weather, climate, air quality at both inter and intraurban scales for each and all cities worldwide.

**Section 2** describes in several subsections, specific details of each of the various methodology and approaches employed to generate this global LCZ map. In general, the approach pertinent to each aspect is described in detail, thoroughly, albeit, many supporting technical details were provided in cited references.

Section 2.1 Training Areas (TAs) This subsection describes the methods to generate this Global LCZ map. It is based on incorporating TAs of LCZs from various sources, mainly from (a) archived community generated TAs representing hundreds of different cities around the world and additionally (b) a special set prepared for another hundred or more other Regions of Interest (ROIs) cities around the world and (c) from TA samples generated by a unique LCZ Generator employed earlier for regional mapping projects (reference cited). The archived TAs required a curation effort to rectify the issue of unevenness in the quality and physical size of TAs submitted into the WUDAPT archives. Their approach adopted in the curation processing is logical, and sound towards assuring uniformity and consistency in quality of the TAs. It clearly builds upon insights and experience gained in prior efforts including the HUMINEX project, implementing such approach contributing to the successful generating of regional to continental scale LCZ maps. The

efforts described in this section extends the approach used for the regional maps to assure a uniformity in the quality of this global product.

_Section 2.2_ describes special treatments to extend the handling of the added and supplementary Earth Observations (EO) needed as inputs for the Global LCZ supervised random forest classifier from the regional mapping stage. Here updates or additions to the original 33 Global Earth Observation were incorporated to the LCZ Generator.

_Section 2.3: Classification schemes_ This section describes what the authors call a Lightweight Global Random Forest model based on various pixel-based mapping methods; however, upscaling such methods to the global product was apparently not straightforward. For this, the default LCZ Generator from earlier studies required significant modifications. This was apparently a huge classification challenge (given >$10^6$ labelled TA and other inputs); it was facilitated by incorporating this dual sequential pathways approach, another innovative advancement.

The important QA assessment builds upon their recent continental scale LCZ mapping efforts; its procedures are based on (a) five (5) traditional accuracy metrics (Section 2.4.1) and (b) incorporating a novel but indirect thematic benchmarking (Section 2.4.2) involving comparing mapped outcomes of several urban canopy parameters ( % built, % impervious surface and sum of built and impervious total plan area, building heights and AH (anthropogenic heating ) associated with each LCZ class with other sets of global and open source databases reflecting urban form and functions. The level of comparability provided a relative qualitative assurance measure of the outcomes of UCPs associated with the LCZ maps. Clearly, the success of these relative outcomes varied for the different UCP analyzed (Fig 10).  In this regard, future effort associated with other independent means including outcomes of UCPs generated by WUDAPT Level 1 and 2 approaches described in Ching et al, (2019) will be helpful, going forward.

_Section 3: Results_. The Global LCZ map is shown in Fig 4. While the 7 non-urban LCZ classes are not as discriminating in the number of classes as in other mapping schemes, the major value is that all urban areas herein are discriminated into the 10 universally based LCZ scheme, a product consistent for all urban areas in the world, e.g., Figure 5-7, are examples of zoomed LCZ maps for various cities extracted from this Global LCZ maps. It was noted that these sets of cities display and support earlier observations that that each and all cities has its uniquely characteristic LCZ signature (or fingerprint).  This is a feature that was apparent and evident in the UCPs generated for the NUDAPT project (Ching et al, 2010), and for LCZ and UCP maps for individual cities studies and from the recent regional LCZ mappings.  From such observations, it is probably reasonable to infer that all cities in the world have unique LCZ fingerprints, and by extension, a commensurate unique set of UCPs. This is an important consideration as it provides the rationale and bases of conducting fit-for-purpose intraurban modeling studies based on the Global LCZ map applicable to each and every city in the world. Support for the contention is expressed in Section 4 indicating how this Global LCZ map can serve earth system modeling and urban scale environmental science, and extend its utility to intraurban scale. However, as noted in the paragraph beginning at line 492, much more need to be done for full effectiveness especially at the intraurban scale in this regard. The UCPs provision in LCZ is currently manifested in lookup tables of ranges of values in the UCPs of each LCZ class. Remedies include path forward innovative cyber-based approaches are currently underway to generate block scale gridded UCPs of both form and function parameters in WUDAPT Level 1 and 2 staged efforts already referred to in the reference list (Ching et al. 2019) to complement the global LCZ mapping efforts.

_Section 4_: this Section, the article highlights and discuss the attributes and impact of the Global LCZ product.  Herein, the significance in terms of objectives, and potential impacts and caveats of this study are explored, Since this global map has just been completed, the discussion refers to the

Global LCZ maps support to wide range of potential modeling applications, some already underway, in concert with the WUDAPT perspecitives.

*Conclusion section*: The results of this Global LCZ map is an important and significant achievement culminating from the collaborative efforts of many activities and voluntary contributions from the urban climate and multidisciplinary science communities evolving and improving after over nearly a decade of efforts. The advances and contribution by this team has been impressive moving the LCZ framework from prototypic city specific mapping to the creation of regional maps and culminating in this impressive global product. Given the widespread and rapidly growing literature on LCZs we can anticipate much interest in this product.  For a whole host of reasons, including projections of climate change impacts to enhancing weather extremes, to urbanization dynamics from increased population, the LCZ paradigm, and the various levels of coverage, and certainly, with this Global product provides an important and significant approach towards supporting science-based tools for myriad and wide ranges of modeling application and studies supporting local to international policies that addresses climate change issues.

*Data availability*: In the last section, this map is available via link provided. While it was not mentioned, I would recommend the authors consider making the map and accessibility to the pixel generated LCZ in the WUDAPT Portal

Please see the response below.

**Suggestions on specific points:** The following are a list of relatively minor issues and concern indicated below.

Line 120-121.This sentence bears a burden of explanation; It will be necessary to provide objective measures and criteria to establish the objective quality indicators of the RUB produced TA vs other ARC sets.

More details on the filtering procedures for the different sets of TAs are provided in lines 126 to 136. In general, it is important to realise that all TA polygons are used, from the three available sources: RUB, ARC and GEN. Irrespective of the source of the samples, they are subjected to general filters, eg. retaining only the submissions with OA > 50%, or removing too small or too complex TA polygons.

Yet only in one filter a prioritisation is applied. That is when submissions from multiple sources cover the same city. In this case we prioritise RUB > ARC > GEN (as indicated on Line 129).  This priority is chosen because 1) it follows the order of the average OA across these sources (RUB: 73%, ARC: 70% and GEN: 65% - after filtering for those with OA > 50%), 2)  RUB samples are given priority as these are produced in a more controlled setting, and 3) also ARC samples are a product of a more controlled setting, as most of these TA sets are the outcome of a peer-review paper and/or quality-controlled by WUDAPT-experts as part of the manual quality procedure in the original WUDAPT portal.

Note that the controlled setting for the RUB samples is further explained in the answer to question 1 of reviewer 1: " Please define what a well-trained student is".

Line 123, page 6.Revise, eliminate term "old" in old WUDAPT Portal. This eliminates the need to explain or differentiate the progression of Portal versions.

We appreciate this feedback. We changed "old" to "original" as we do want to make a distinction between the first portal (that is currently no longer maintained nor updated, as indicated on https://www.wudapt.org/the-wudapt-portal/), and its successor, the LCZ Generator.

Line 131, "Third" used here is really "Fourth" as "Third is already used in line 129

The numbering of these sentences has been adjusted.

Line 189:To better understand and appreciate the value of the Lightweight global random forest model, herein, as regards the probability layer, please discuss the takeaway points of the meaning of a high vs low probability.

The lightweight Random Forest model was introduced for computing efficiency since the entire dataset was too large to train one classifier (even with the tremendous computing resources offered by Google Earth Engine). Thus the ensemble-based decision making principle of Random Forest was adapted using an outer split to reduce the problem to several smaller problems, i.e. 50 classification models based on a subset of the data. The effect on the accuracy is unknown but considered to be minor, since the procedure is similar to what RF does internally anyway (with different parameters, sensitivity which was tested in pathway 1), just with our approach only 15 % of the memory is needed for each run. As added value the results from the 50 runs can be interpreted as a classification probability (see also rebuttal Review #1). A high classification probability means that most of the 50 RF models agree (i.e. class is chosen independent of training subset). A low classification probability means that the RF models largely disagree (i.e. results strongly depend on the training subset).

The role of the classification probability layer is described in Section 3.2, Figure 9, and a new Figure H1. In short, this section describes that "*all classification probabilities per LCZ class are in line with the global values, demonstrating the universality of the LCZ typology and the robustness of the classifiers and input features across the urban ecoregions*". It also indicates areas where more work is needed, such as the mapping of LCZ class 7, which is indicated as one of the caveats in the Discussion section.

Line 246 Functional Urban Area (FUA) are indicated by a reference (Schiavina et al. 2019); given the important role of FUAs in discussions in the remaining text it would be highly useful to introduce the key aspects of this FUA framework as in the Section 2.3 and 3.2.
Please note that in-depth information on FUAs is already provided in Appendix C, which will be part of the main manuscript and will thus not be provided as supplementary information:
*Functional urban areas (FUAs), as defined by the Organisation for Economic Co-operation and Development (OECD) and the European Union, are sets of contiguous local (administrative) units composed of a city and its surrounding, less densely populated local units that are part of the city's labour market (commuting zone). As such, these units not only offer the opportunity to evaluate more densely built city centres, yet also their sparsely built or natural neighbouring landscapes.*

This Appendix is explicitly referenced upon the first usage of FUAs. In order not to clutter the main manuscript with too many details, we have decided to keep this information in this Appendix.

Line 433 pg 24. Clarification of introducing the Term "scaling laws"
This paragraph has been adjusted to better explain the idea of "scaling".

Line 503: Consider elaborating on introducing the term "subclass scheme"
Some information is added to clarify this, yet more importantly the paper from Stewart and Oke (2012) is explicitly mentioned, since it provides much more details in the LCZ subclasses.

Explain "Morphological Gaussian Filter"
In general, many parts of this work built upon previous results, that are described in previously published works. We believe it is not required to repeatedly explain underlying procedures, as long as the original papers describing those are properly referenced. In the case of the morphological gaussian filter, this is also the case, rereferring to Demuzere et al. (2020a).

Highly recommend the Global LCZ Map and data results be incorporated into the WUDAPT Portal.

Since the original WUDAPT portal is no longer maintained nor updated, we do not plan to put the Global LCZ map data there. We do however foresee a number of outlets that will allow interested users to access the data:
1. The official Zenodo data archive: https://zenodo.org/record/6364594. Even though this manuscript is not officially accepted yet, the data has been seen and downloaded 600+ times already [last accessed July 11 2022].
2. Interactive viewer on the LCZ Generator: we are currently designing another tab on the LCZ Generator page that will display the global map in an interactive manner.
3. GIS and other tools: Since we had to develop tiles for 2., these will also be accessible by other software such as QGIS and similar, that support xyz tile servers.
4. Earth Engine: once the manuscript is accepted, the data will be released in the official Earth Engine data repository, as described here: https://developers.google.com/earth-engine/help_dataset_description.

Page 51, Last line of caption to Figure F1; add "detail in" the Table F1.
Good catch! "detail' is added.

**Summary**:  The effort described in this article to creating this Global LCZ map is impressive. This product represents an achievement, culminating a decade of activity and efforts by the urban community through WUDAPT towards acquiring urban canopy layer data for models that provide a means for addressing climate change and urbanization issues for local to regional to global scales. Its paradigm incorporates the LCZ typology thus providing a unified and consistent basis for generating intraurban form and function type data paradigm (WUDAPT Level 0) for the urban canopy layer.  This Global LCZ map supports the observation from earlier city specific studies and regional maps, that each city LCZ map signature (e.g. fingerprint) is unique. This "feature" provides a rationale for supporting a wide range of earth systems modeling, applications, and urban scale environmental science, e.g., urban modeling applications in which intraurban scale weather forecasting and assessments can be made based on implementing urban boundary layer parameterizations in models with universally consistent intraurban urban canopy descriptions unique to each and every urban area in the world. The rationale, requisite technical issues to the innovative approach toward generating this global map, the approach and results were well articulated and fully documented. The point to future work advancements some alluded to in the caveats provided of the current LCZ paradigm for balance. In particular, its incorporation into WRF for example through an improved WUDAPT to WRF link is underway as well as efforts along the lines of WUDAPT level 1 and 2 (Ching et al. (2019) will be to provide a pathway towards introducing refined city specific block scale gridded UCPs in future updates.
Thank you Dr. Ching for the supportive words!